# NeuroSketch: An Effective Framework for Neural Decoding via Systematic Architectural Optimization

## Abstract

Neural decoding, a critical component of Brain-Computer Interface (BCI), has recently attracted increasing research interest. Previous research has focused on leveraging signal processing and deep learning methods to enhance neural decoding performance. However, the in-depth exploration of model architectures remains underexplored, despite its proven effectiveness in other tasks such as energy forecasting and image classification. In this study, we propose NeuroSketch, an effective framework for neural decoding via systematic architecture optimization. Starting with the basic architecture study, we find that CNN-2D outperforms other architectures in neural decoding tasks and explore its effectiveness from temporal and spatial perspectives. Building on this, we optimize the architecture from macro- to micro-level, achieving improvements in performance at each step. The exploration process and model validations take over 5,000 experiments spanning three distinct modalities (visual, auditory, and speech), three types of brain signals (EEG, SEEG, and ECoG), and eight diverse decoding tasks. Experimental results indicate that NeuroSketch achieves state-of-the-art (SOTA) performance across all evaluated datasets, positioning it as a powerful tool for neural decoding.

## 1 Introduction

Brain-Computer Interface (BCI) technology aims to revolutionize interaction methods by establishing a direct link between thought and action, enabling more efficient communication (Maiseli et al., 2023). Neural decoding (Van Gerven et al., 2019) plays a critical role in this process, as it involves inferring external stimuli, cognitive states, or intentions from brain signals. These signals are typically recorded using methods such as electroencephalography (EEG) (Teplan, 2002), stereoelectroencephalography (SEEG) (Talairach, 1974), and electrocorticography (ECoG) (Shenoy et al., 2007), which can be classified as non-invasive and invasive techniques. EEG, a non-invasive method, records electrical activity along the scalp and is widely used in both research (Zhang et al., 2021; Altaheri et al., 2023; Rahman et al., 2021) and clinical settings (Pani et al., 2022; Saminu et al., 2023) due to its practicality and low cost. In contrast, invasive methods like SEEG and ECoG capture signals from deeper brain structures, offering higher temporal and spatial resolution compared to non-invasive methods (Chaddad et al., 2023), and enabling more precise identification of neural activity linked to specific cognitive functions.

In neural decoding, recorded brain signals are characterized by transient temporal dynamics (King and Dehaene, 2014) and spatial locality (Mahjoory et al., 2024), representing unique modeling demands. Traditional time series data, such as weather, electricity consumption, and traffic flow, are sampled at a relatively low frequency (*e.g.,* minutes or hours) and often exhibit significant periodicity or trends. However, brain signals require higher temporal resolution to capture the brain's rapidly changing states. Even within brain signals, their characteristics can differ significantly across various scenarios. From a temporal perspective, brain signals typically contain crucial information within much shorter time frames in neural decoding tasks compared to other scenarios, such as sleep staging. In sleep staging tasks, brain signals are recorded overnight, with a labeling resolution of 30 seconds for each sleep stage (Phan and Mikkelsen, 2022). In contrast, in the Rapid Serial Visual Presentation (RSVP) paradigm for image stimulus decoding, relevant information is encoded within just a few hundred milliseconds (Butts et al., 2007). From a spatial perspective, task-driven neural

activity in decoding paradigms differ fundamentally from those of pathological neural disorders. Pathological conditions such as generalized epilepsy exhibit widespread brain activity, resulting in non-specific neural activations (Stafstrom and Carmant, 2015). In neural decoding tasks, however, external stimuli evoke localized activations that are typically restricted to functionally specialized regions and recorded by spatially proximate electrodes. For example, speech stimuli predominantly activate the left inferior frontal gyrus (Leonard et al., 2024), while visual stimuli engage the visual cortex (Grill-Spector and Malach, 2004).

Previous work on neural decoding can be broadly categorized into two approaches. The first is signal processing (Peksa and Mamchur, 2023), which focuses on improving the signal-to-noise ratio and extracting task-relevant features (Wittevrongel et al., 2020; Proix et al., 2022; Safi and Safi, 2021). These methods rely on manually defined features, which are time-consuming and highly empirical. The second approach involves deep learning methods (Angrick et al., 2019; Makin et al., 2020; Wilson et al., 2020; Willett et al., 2023; Metzger et al., 2023; Zheng et al., 2024). However, current approaches largely overlook in-depth exploration and modeling of the temporal dynamics and spatial locality inherent to neural decoding. Therefore, our work aims to explore a framework more suited to neural decoding tasks through systematic architectural optimization. Such optimizations have already yielded significant benefits in other tasks (Yu et al., 2022; 2023; Luo and Wang, 2024; Wang et al., 2025a). For instance, in energy forecasting tasks, ModernTCN (Luo and Wang, 2024) reduces MSE by 13.9% on ETTm2 (Zhou et al., 2021) compared to convolution-based models by using large kernel convolutions and a ConvFFN module. In image classification tasks, ConvNeXt (Liu et al., 2022) surpasses ResNet-50 (He et al., 2016) on ImageNet (Deng et al., 2009) through optimizations in stage compute ratio and convolutional methods.

Our study is guided by two core questions that aim to systematically investigate how to design an effective framework for neural decoding:

*Q1: Which basic model architecture is best suited to capture temporal and spatial patterns in brain signals?*

*Q2: Based on an appropriate architecture, how can we improve neural decoding performance through macro-to-micro architectural optimization?*

As shown in Figure 1, by answering the above questions, we consistently improve the decoding performance at each optimization step. First, we study 9 basic architectures (CNN (He et al., 2016), GRU (Chung et al., 2014), Transformer (Vaswani et al., 2017), and their variants) and find that CNN-2D outperforms the others. The results are coherent with the characteristics of neural decoding signals, since CNN-2D is suitable for capturing localized patterns from both temporal and spatial dimensions. Based on CNN-2D, we design the entire framework us-

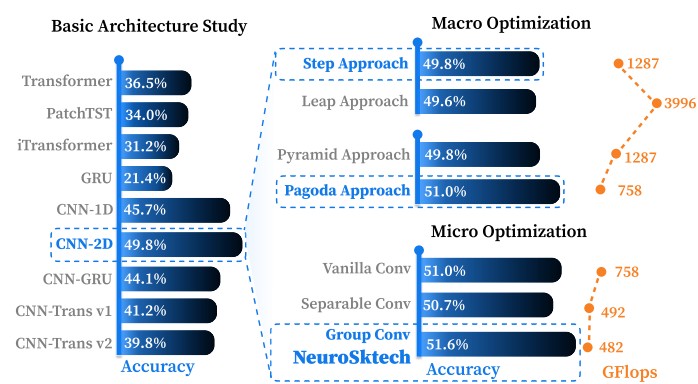

Figure 1: **Roadmap of architectural optimization.**

ing a macro-to-micro paradigm, analyzing the latent space transformation during forward propagation and optimizing the calculation of micro components. Through the above step-by-step optimization process, we propose NeuroSketch, an effective framework for neural decoding. NeuroSketch aligns with the temporal and spatial characteristics of brain signals in neural decoding tasks, achieving state-of-the-art performance across eight neural decoding tasks, which can serve as a useful tool for future research and applications in neuroscience.

In summary, the main contributions of this research are as follows:

1. We propose NeuroSketch, an effective framework that aligns with the temporal and spatial characteristics of brain signals for neural decoding tasks.

2. From the technical perspective, we systematically reexamine the design space and consider several critical components. Starting with exploring basic architectures, we progressively investigate architectural design principles from macro- to micro-level.

3. From the experimental perspective, we validate NeuroSketch through over 5,000 experiments across eight neural decoding tasks related to vision, hearing, and speech, recorded by different types of brain signals. The experimental results demonstrate that NeuroSketch achieves state-of-the-art decoding performance across all evaluated datasets.

## 2 ROADMAP TOWARDS AN EFFECTIVE FRAMEWORK

In this section, we provide a trajectory that outlines the steps towards achieving an effective framework. The roadmap of our study is essentially focused on answering two key questions mentioned in Section 1. To begin with, Section 2.1 describes the experimental setup. Then, Section 2.2 addresses Question 1 through systematic analyses of the basic model architecture, considering both temporal and spatial perspectives. Building upon the appropriate architecture, Section 2.3 investigates the macro-level optimization aspect of Question 2, focusing on the latent space transformation. Subsequently, Section 2.4 explores the micro-level optimization aspect of Question 2, focusing on computation optimization. Finally, Section 2.5 integrates the above findings and introduces NeuroSketch.

### 2.1 EXPERIMENT SETUP

To rigorously evaluate our exploration, we conduct nearly 2,000 experiments on eight neural decoding tasks, spanning three major categories—speech, visual, and auditory decoding—and three types of brain signals—EEG, SEEG, and ECoG. The overview of each dataset is provided in Table 1, with further details in Appendix C. Since the network complexity closely correlates with final performance, the model's parameter size is kept around 30M by adjusting the model depth and embedding dimension during the experimental process. Detailed parameter settings and training setup can be found in Appendix D.1 and Appendix E.

Table 1: **Overview of the datasets used in our experiments.** We evaluate on six datasets in total. For *Chisco* and *OpenMIIR*, we partition them by task, yielding eight distinct tasks overall.

| Categories | Datasets | Tasks | Signal Type |
|---|---|---|---|
| Speech | Du-IN (Zheng et al., 2024) | Loud Reading | SEEG |
|  | Chisco-R (Zhang et al., 2024) | Silent Reading | EEG |
|  | Chisco-I (Zhang et al., 2024) | Speech Imagination | EEG |
| Visual | FacesHouses (Miller, 2019) | Binary Image Decoding | ECoG |
|  | ThingsEEG (Gifford et al., 2022) | Multi-Class Image Decoding | EEG |
|  | SEED-DV (Liu et al., 2024a) | Video Decoding | EEG |
| Auditory | OpenMIIR-P (Stober et al., 2015) | Music Perception | EEG |
|  | OpenMIIR-I (Stober et al., 2015) | Music Imagination | EEG |

### 2.2 BASIC ARCHITECTURE STUDY

To address Question 1, we investigate the performance of nine commonly used architectures in neural decoding tasks, selected from four categories: Convolutional Neural Networks (CNNs), Gated Recurrent Units (GRUs), Transformers, and their hybrids. The input of these models can be represented as $\mathbf{X} \in \mathbb{R}^{B \times C \times L}$, where $B$ denotes the batch size, $C$ denotes the number of channels, and $L$ denotes the number of time steps. We study two types of CNNs: CNN-1D and CNN-2D. CNN-1D applies 1D convolutional filters across the temporal dimension, with each filter combining features from all channels. CNN-2D reshapes the input $\mathbf{X}$ to $\mathbf{X}' \in \mathbb{R}^{B \times 1 \times C \times L}$ and applies 2D convolutional filters across both the temporal and channel dimensions. In the case of Vanilla GRU, the model sequentially processes the input $\mathbf{X}$ across the temporal dimension, updating its hidden state at each time step using both the current input and the previous state. In contrast, the Transformer processes the input $\mathbf{X}$ in parallel using self-attention to capture dependencies across all time steps. We also consider two Transformer variants: PatchTST(Nie et al., 2023) and iTransformer(Liu et al.,

2024b). PatchTST divides the input $\mathbf{X}$ along the temporal dimension into patches of length $P$ and stride $S$, generating a sequence of $N$ patches, where $N = \lfloor \frac{L-P}{S} \rfloor + 2$. Then, it processes the patched input $\mathbf{X}_p \in \mathbb{R}^{B \times C \times N \times P}$ using Transformer layers to capture the relationships between different patches while maintaining channel independence. In contrast, iTransformer treats channels as sequence elements and uses self-attention to capture interactions across channels. In addition to individual models, we explore hybrid architectures that combine multiple approaches, including CNN-GRU and two CNN-Transformer variants. One variant uses CNN for feature extraction followed by Transformer layers (Song et al., 2022), while the other integrates both CNN and Transformer within each module to jointly capture temporal information at different levels (Kim et al., 2022). CNN-GRU, similar to the first CNN-Transformer variant, employs CNN for initial feature extraction, with the GRU component then processing the sequential data to capture temporal dependencies.

Table 2: **Results of the basic architecture study and subsequent analyses.** v1 and v2 refer to the two hybrid methods discussed above. The ratio shown in the middle of the table (*e.g.,* 4:1) indicates the proportion of CNN to Transformer layers in each hybrid model. The best results are in **bold** and the second best are underlined.

| Models \ Datasets | Du-IN Acc | Du-IN F1 | SEED-DV Acc | SEED-DV F1 | ThingsEEG Acc | ThingsEEG F1 | FacesHouses Acc | FacesHouses F1 | OpenMIIR-P Acc | OpenMIIR-P F1 | OpenMIIR-I Acc | OpenMIIR-I F1 | Chisco-R Acc | Chisco-R F1 | Chisco-I Acc | Chisco-I F1 |
|---|---|---|---|---|---|---|---|---|---|---|---|---|---|---|---|---|
| *Basic Architecture Study* | | | | | | | | | | | | | | | | |
| CNN-1D | .451 | .446 | .061 | .054 | **.202** | **.191** | .799 | .796 | .973 | .973 | .968 | .968 | .104 | .071 | .100 | .073 |
| **CNN-2D** | **.647** | **.641** | .063 | .058 | .184 | .177 | **.886** | **.885** | **.981** | **.980** | **.982** | **.982** | **.127** | **.113** | .114 | **.095** |
| GRU | .353 | .343 | .025 | .001 | .040 | .037 | .774 | .773 | .146 | .046 | .136 | .039 | .123 | .091 | .117 | .086 |
| Transformer | .085 | .080 | .042 | .028 | .005 | .000 | .795 | .794 | .953 | .952 | .911 | .910 | .069 | .015 | .057 | .012 |
| PatchTST | .060 | .051 | .038 | .029 | .006 | .001 | .845 | .838 | .906 | .904 | .766 | .760 | .050 | .005 | .050 | .005 |
| iTransformer | .097 | .085 | .029 | .005 | .005 | .000 | .730 | .728 | .729 | .698 | .780 | .774 | .064 | .013 | .061 | .014 |
| CNN-GRU | .416 | .409 | .057 | .049 | .172 | .166 | .803 | .801 | .957 | .954 | .943 | .942 | .092 | .064 | .087 | .063 |
| CNN-Trans v1 | .333 | .326 | .067 | .060 | .046 | .027 | .752 | .751 | .948 | .947 | .945 | .945 | .106 | .061 | .100 | .057 |
| CNN-Trans v2 | .282 | .268 | .050 | .045 | .022 | .010 | .754 | .752 | .952 | .950 | .930 | .930 | .097 | .050 | .093 | .059 |
| *Ratio of CNN and Transformer Layers in Hybrid Architectures* | | | | | | | | | | | | | | | | |
| **CNN-Trans v1-4:1** | **.333** | **.326** | .067 | **.060** | **.046** | **.027** | **.752** | **.751** | **.948** | **.947** | **.945** | **.945** | **.106** | **.061** | **.100** | **.057** |
| CNN-Trans v1-3:2 | .115 | .091 | **.069** | .057 | .010 | .001 | .696 | .694 | .904 | .901 | .916 | .914 | .094 | .036 | .090 | .041 |
| CNN-Trans v1-2:3 | .060 | .032 | .041 | .021 | .005 | .000 | .699 | .697 | .833 | .828 | .867 | .866 | .088 | .030 | .085 | .031 |
| CNN-Trans v1-1:4 | .036 | .013 | .030 | .008 | .005 | .000 | .684 | .674 | .709 | .687 | .764 | .760 | .079 | .018 | .076 | .019 |
| **CNN-Trans v2-4:1** | **.282** | **.268** | .050 | .045 | **.022** | **.010** | **.754** | **.752** | **.952** | **.950** | **.930** | **.930** | **.097** | .050 | **.093** | **.059** |
| CNN-Trans v2-3:2 | .211 | .192 | **.061** | **.054** | .024 | .009 | .719 | .713 | .946 | .944 | .882 | .880 | .097 | **.057** | .087 | .051 |
| CNN-Trans v2-2:3 | .141 | .124 | .059 | .046 | .011 | .002 | .710 | .707 | .882 | .881 | .893 | .892 | .088 | .050 | .092 | .044 |
| CNN-Trans v2-1:4 | .047 | .033 | .035 | .019 | .008 | .001 | .722 | .720 | .822 | .814 | .839 | .837 | .094 | .047 | .084 | .041 |
| *New Patch Method for Transformers and GRUs* | | | | | | | | | | | | | | | | |
| Transformer | .085 | .080 | **.042** | .028 | .005 | .000 | .795 | .794 | **.953** | **.952** | .911 | .910 | .069 | .015 | .057 | .012 |
| PatchTST | .060 | .051 | .038 | **.029** | .006 | .001 | **.845** | **.838** | .906 | .904 | .766 | .760 | .050 | .005 | .050 | .005 |
| iTransformer | .097 | .085 | .029 | .005 | .005 | .000 | .730 | .728 | .729 | .698 | .780 | .774 | .064 | .013 | .061 | .014 |
| **Ours** | **.234** | **.226** | .033 | .027 | .005 | .000 | .799 | .799 | .922 | .919 | .843 | .839 | **.098** | **.046** | **.090** | **.066** |
| GRU | **.353** | **.343** | .025 | .001 | .040 | .037 | .774 | .773 | .146 | .046 | .136 | .039 | **.123** | **.091** | **.117** | **.086** |
| **Ours** | .351 | .342 | **.033** | **.026** | **.074** | **.068** | **.817** | **.813** | **.958** | **.956** | **.937** | **.936** | .105 | .071 | .091 | .066 |

Table 2(upper) shows the performance of the 9 basic architectures discussed above. Overall, CNN-based models, which excel in capturing local patterns, outperform hybrid models on most neural decoding tasks. Meanwhile, GRU and Transformer-based models, which are better suited for modeling long-range dependencies, exhibit the lowest performance. This result aligns with the **transient temporal dynamics** of brain signals in neural decoding tasks, suggesting that short-range temporal information is more effective for neural decoding. In CNN-based models, CNN-2D, which extracts local channel information using convolutional kernels, outperforms CNN-1D, which fuses all channel information through addition. This performance difference aligns with the **spatial locality** observed in neural decoding tasks. Surprisingly, the Transformer-based models perform poorly on certain datasets, particularly the ThingsEEG dataset, where their best accuracy of 0.6% is only marginally above the chance level of 0.5%. In order to find out the underlying reasons for the performance differences, we conduct further analyses from both temporal and spatial perspectives.

From a temporal perspective, we investigate how short- and long-term temporal information affects the model performance. We control the short- and long-term information through adjusting the ratio of CNN and Transformer layers in two hybrid models. From the results in Table 2(middle), we observe a decline in performance as the proportion of short-term information decreases, demonstrating the effectiveness of short-range information for neural decoding.

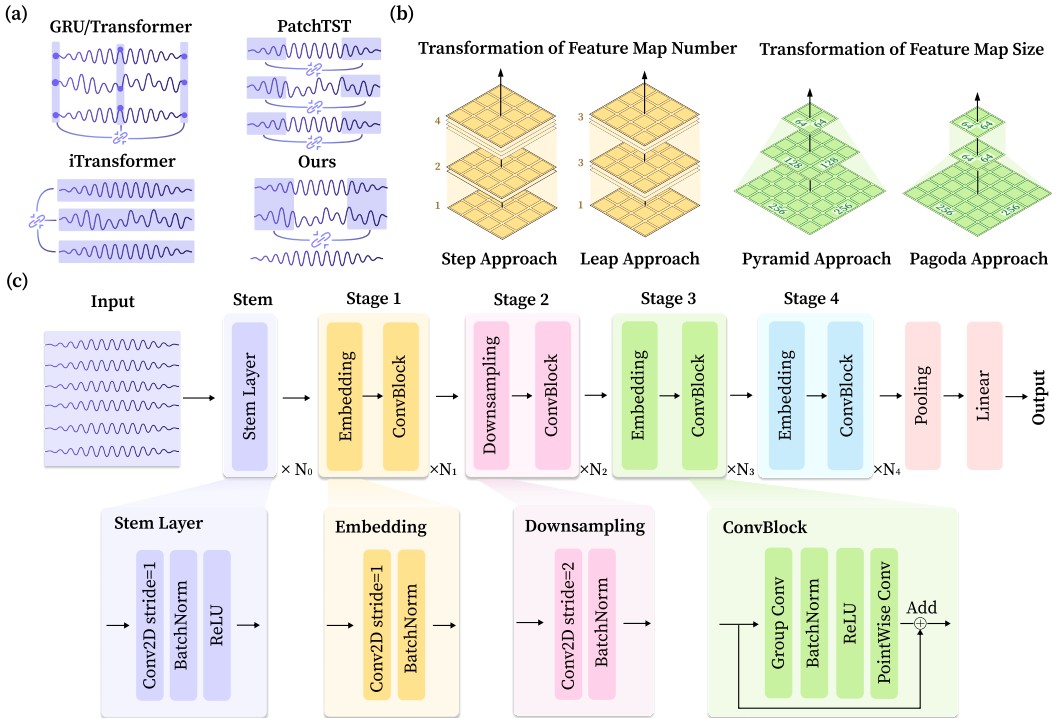

Figure 2: **(a) Comparison of different patch methods.** Vanilla GRU and Transformer treat a single timestamp across all channels as a token; PatchTST treats multiple timestamps of a single channel as a token; iTransformer treats all timestamps of an entire channel as a token. We propose a simple but effective patch method that aggregates information of multiple timestamps and channels. **(b) Comparison of different latent space transformation methods.** Regarding the number of feature maps, the step approach increases them gradually, whereas the leap approach increases them rapidly in the early stages. In terms of their size, the pyramid approach decreases them progressively, while the pagoda approach decreases them quickly in the early stages. **(c) The overall architecture of NeuroSketch.**

From a spatial perspective, based on the effectiveness of CNN-2D, we can similarly apply this finding to other models, such as GRU and Transformer, to further validate its applicability. We propose a simple but effective patch method to capture local channel information in Transformers and GRUs. As shown in Figure 2(a), the input sequence is first divided into patches using a sliding window approach. Unlike the patch methods proposed in PatchTST and iTransformer, which treat channels independently or focus on global channel information, we reshape the patches by concatenating the channel and patch dimensions to preserve the local channel relationships within the same time window. The results in Table 2(lower) demonstrate that our method achieves highly competitive performance compared to others. By incorporating our token embedding method, the accuracy of the Transformer on the Du-IN dataset improved significantly from 8.5% to 23.4%. Additionally, the accuracy of the GRU on the OpenMIIR dataset's perception and imagination tasks increased dramatically from the chance level to 95.8% and 95.7%, respectively, further demonstrating the effectiveness of spatial locality for neural decoding.

*From now on, we will use CNN-2D as our basic architecture.*

## 2.3 Latent Space Transformation

To address the macro optimization aspect of Question 2, we study the latent space transformation during the forward propagation. For CNN-2D, its latent space representation refers to $\mathbf{S} \in \mathbb{R}^{C \times H \times W}$, where $C$ denotes the number of feature maps, $H$ and $W$ denote the height and width of feature maps, respectively. During forward propagation, the network increases the number of feature maps to capture higher-level information, while reducing the size of feature maps to alleviate redundancy and improve computational efficiency. Building on this general overview, we will investigate the transformation of the number ($C$) and the size ($H$ and $W$) of feature maps separately.

Regarding the number of feature maps, the appropriate strategy to increase them is a critical aspect of neural network design, and this can be achieved through two widely adopted approaches: **step approach** and **leap approach**. As shown in Figure 2(b), The step approach gradually increases the number of feature maps, allowing feature extraction to advance from low-level to high-level in a systematic manner. This progression enables the network to concentrate on varying levels of features effectively. The leap approach takes a more aggressive strategy to rapidly increase the number of feature maps to the embedding dimension in the early stages, making the model focus on refining and extracting higher-level features.

Table 3: **Results of the latent space transformation and the optimization of calculation.** Lower GFLOPs indicates lower computational cost. The best results are in **bold**.

| Setting | GFLOPs | Du-IN | | SEED-DV | | ThingsEEG | | FacesHouses | | OpenMIIR-P | | OpenMIIR-I | | Chisco-R | | Chisco-I | |
|---|---|---|---|---|---|---|---|---|---|---|---|---|---|---|---|---|---|
| | | Acc | F1 | Acc | F1 | Acc | F1 | Acc | F1 | Acc | F1 | Acc | F1 | Acc | F1 | Acc | F1 |
| *Macro Study: Transformation of the Feature Map Number* | | | | | | | | | | | | | | | | | |
| Leap Approach | 3996 | .635 | .630 | **.064** | .058 | **.216** | **.206** | .857 | .856 | .980 | .980 | .976 | .976 | .126 | .109 | **.115** | .093 |
| **Step Approach** | **1287** | **.647** | **.641** | .063 | **.058** | .184 | .177 | **.886** | **.885** | **.981** | **.980** | **.982** | **.982** | **.127** | **.113** | .114 | **.095** |
| *Macro Study: Transformation of the Feature Map Size* | | | | | | | | | | | | | | | | | |
| Pyramid Approach | 1287 | .647 | .641 | **.063** | **.058** | .184 | .177 | .886 | .885 | **.981** | **.980** | .982 | .982 | .127 | .113 | .114 | .095 |
| **Pagoda Approach** | **758** | **.701** | **.698** | .059 | .053 | **.204** | **.197** | **.911** | **.910** | .973 | .973 | **.986** | **.985** | **.128** | **.116** | **.116** | **.096** |
| *Micro Study* | | | | | | | | | | | | | | | | | |
| Vanilla Conv | 758 | .701 | .698 | .059 | .053 | .204 | .197 | .911 | .910 | .973 | .973 | .986 | .985 | .128 | **.116** | .116 | .096 |
| Seperable Conv | 492 | .704 | .699 | .061 | .051 | .181 | .176 | .915 | .915 | .979 | .979 | .985 | .985 | .118 | .107 | .111 | .098 |
| **Group Conv** | **482** | **.707** | **.704** | **.069** | **.061** | **.207** | **.200** | **.920** | **.920** | **.983** | **.983** | **.990** | **.990** | **.129** | .112 | **.120** | **.103** |

Table 3(upper) compares the performance and computational cost between the step and leap approach. Surprisingly, the step approach achieves highly competitive performance compared to the leap approach, while requiring 67.8% fewer FLOPs. For instance, the step approach attains an accuracy of 88.6% on the FacesHouses dataset, to achieve similar accuracy, the leap approach requires three times more computational cost, while only reaching 85.7% accuracy. This indicates that neural decoding does not rely solely on high-level neural representations, and that multi-scale feature extraction is often more effective and efficient.

*From now on, we will use the step approach to transform the number of feature maps.*

In the forward propagation process, the transformation of the feature map size is achieved through downsampling. Therefore, we further investigate how downsampling should be distributed throughout the network. Suppose downsampling is applied only in the final layers. In that case, earlier layers must process high-resolution feature maps, which significantly increases computational complexity and undermines the original purpose of downsampling to improve computational efficiency. Therefore, we study two common strategies: **pyramid approach** and **pagoda approach**. As shown in Figure 2(b), the pyramid approach distributes the downsampling process evenly across the entire network, gradually reducing the size of the feature maps. This pyramid-like structure enables the network to retain more detailed features at the cost of increased computational complexity. The pagoda approach takes a more aggressive strategy by concentrating the downsampling module in the early stages of the network. This leads to a significant reduction in feature map size early on, resulting in lower computational costs. Table 3(middle) compares the performance and computational cost between the pyramid and pagoda approach. The pagoda approach achieves slightly better accuracy than the pyramid approach, requiring 41.1% fewer FLOPs. Specifically, on the Du-IN dataset, the pagoda approach attains 70.1% accuracy, whereas the pyramid approach only reaches

64.7% with a higher computational cost. Additionally, the pagoda approach improves the accuracy by 2.0% to 20.4% on the ThingsEEG dataset, surpassing CNN-1D's accuracy of 20.2%, making it the best-performing architecture. These results demonstrate that excessively extracting detailed information from the original signal is unnecessary, as brain signals typically have low signal-to-noise ratios. By applying downsampling early in the network, we can enhance computational efficiency while improving performance, allowing the model to focus on the most relevant features and reduce the impact of noise.

*From now on, we will use the pagoda approach to transform the size of feature maps.*

### 2.4 Optimization of Calculation

To further explore the micro optimization aspect of Question 2, we investigate the core calculation method of CNNs, the convolution operation. Given $\mathbf{X}_{\text{in}} \in \mathbb{R}^{C_{\text{in}} \times H_{\text{in}} \times W_{\text{in}}}$ as the $C_{\text{in}}$ channels input of height $H_{\text{in}}$ and width $W_{\text{in}}$, the convolution kernel $\mathbf{F} \in \mathbb{R}^{C_{\text{out}} \times C_{\text{in}} \times U \times V}$ of height $U$ and width $V$ slides across the input to compute the output $\mathbf{X}_{\text{out}} \in \mathbb{R}^{C_{\text{out}} \times H_{\text{out}} \times W_{\text{out}}}$, where $C_{\text{out}}, H_{\text{out}}$ and $W_{\text{out}}$ refer to the output channels, height and width, respectively. The number of parameters involved in this convolution operation is $C_{\text{out}} \times C_{\text{in}} \times U \times V$, representing the learnable weights within the convolution kernel.

In addition to the vanilla convolution, various variants have been proposed to improve computational efficiency and model performance, including group convolution (Ioannou et al., 2017) and separable convolution (Howard, 2017).

**Group convolution.** Group convolution $\mathbf{F}_g \in \mathbb{R}^{C_{\text{out}} \times \frac{C_{\text{in}}}{G} \times U \times V}$ divides the input channels $C_{\text{in}}$ and output channels $C_{\text{out}}$ into $G$ groups. The convolution is then applied separately within each group, limiting the channel interaction and effectively reducing the parameters by a factor of $G$ compared to the vanilla convolution.

**Separable convolution.** Separable convolution breaks down the vanilla 2D convolution into two distinct operations: depthwise convolution $\mathbf{F}_d \in \mathbb{R}^{C_{\text{in}} \times U \times V}$ for spatial dimensions and pointwise convolution $\mathbf{F}_p \in \mathbb{R}^{C_{\text{out}} \times C_{\text{in}} \times 1 \times 1}$ for channel dimension. Depthwise convolution is an extreme case of group convolution, where the number of groups $G = C_{\text{in}} = C_{\text{out}}$. In this case, each output feature map corresponds directly to a specific input feature map. To enable channel interactions, a pointwise convolution is typically applied afterward, where both the kernel height and width are set to 1. In total, the number of parameters in a separable convolution is $\frac{1}{C_{\text{out}}} + \frac{1}{UV}$ of that in vanilla convolution.

Here, we examine how performance changes when replacing the vanilla convolution with separable and group convolution. As shown in Table 3(lower), the group convolution variant achieves slightly better performance compared to other convolution methods while minimizing computational cost. Besides, the separable convolution variant also shows competitive performance with less computational cost than vanilla convolution. For example, with 758 GFLOPs, the vanilla convolution gets 5.9% accuracy on the SEED-DV dataset, while the separable convolution can obtain 6.1% accuracy with 35% fewer GFLOPs. Meanwhile, with nearly identical GFLOPs, the group convolution outperforms the separable convolution, improving accuracy by 0.8% to 6.9%. The experimental results show that grouping input and output channels for convolution does not degrade performance in neural decoding tasks. Channel grouping may be more suitable for aggregating local information while improving computational efficiency.

*We will use group convolution as our core calculation method. This brings us to our final framework, NeuroSketch.*

### 2.5 Put It All Together

After the optimizations mentioned above, we have developed an effective framework, NeuroSketch. The overall architecture is shown in Figure 2(c). Like CNN-2D, NeuroSketch employs a 2D input representation, which is then processed by a stem layer to initiate feature extraction. The following forward propagation process is divided into four stages to progressively capture features from low-level to high-level, as suggested by the step approach. In each stage, the initial component is responsible for increasing the number of feature maps. If this component is a downsampling layer, it

also reduces the spatial resolution of the feature maps by half. Based on the pagoda approach, we allocate the downsampling layers to the second stage to enhance performance and computational efficiency. Subsequently, a convolutional block is applied to extract features, comprising group convolutions, batch normalization, ReLU activation, and pointwise convolution. Following the four stages, the resulting feature is passed through a generalized mean (GeM) pooling layer (Berman et al., 2019) that aggregates features along the channel and temporal dimensions. The pooled representation is then fed into a linear layer for the final classification. Additional details of the architecture and implementation can be found in Appendix D.2.

# 3 EXPERIMENTS

In this section, we present a comprehensive evaluation of NeuroSketch based on more than 3,000 experiments. Section 3.1 introduces the datasets, baselines, and experimental settings, while Section 3.2 reports the main results, including analyses across three different modalities: speech decoding, visual decoding, and auditory decoding.

## 3.1 EXPERIMENT SETUP

We employ the same datasets described in Section 2 for evaluation. The diverse selection of datasets ensures a comprehensive evaluation across different neural decoding tasks. We implement two versions of NeuroSketch with different sizes: NeuroSketch-Base (1.4M parameters) and NeuroSketch-Large (4.2M parameters). Detailed implementation and configuration are provided in Appendix D.2. For baselines, we select representative and recent models from various domains, including time-series models: ModernTCN (Luo and Wang, 2024), MedFormer (Wang et al., 2024); computer vision backbones: ConvFormer (Yu et al., 2023), CAFormer (Yu et al., 2023); well-known brain models: DeepConvNet (Schirrmeister et al., 2017), EEGNet (Lawhern et al., 2018); recent brain models: Conformer (Song et al., 2022), SPaRCNet (Jing et al., 2023); iEEG foundation models: seegnificant (Mentzelopoulos et al., 2024); and EEG foundation models: CBraMod (Wang et al., 2025b). We use the official pretrained weights for all foundation models. More details of each baseline and evaluation setup are introduced in Appendix D.3 and Appendix E.

## 3.2 RESULTS

**Main results.** Overall, NeuroSketch consistently achieves state-of-the-art performance across eight neural decoding tasks. As shown in Figure 3, NeuroSketch outperforms all selected baselines, underscoring its strong capacity to model brain signals across diverse decoding scenarios. In addition, we note that NeuroSketch-Base performs similarly to NeuroSketch-Large on all datasets except Du-IN, where NeuroSketch-Large outperforms NeuroSketch-Base by a substantial margin. We discuss this phenomenon in detail in Appendix F.3.

**Speech decoding.** Table 4 reports the excellent performance of NeuroSketch on the three speech decoding tasks. Notably, on the Du-IN dataset, NeuroSketch-Large surpasses the second-best baseline, ConvFormer, by 65.5% in accuracy, underscoring its strength in extracting discriminative features from neural signals. As an iEEG foundation model, seegnificant performs poorly on Du-IN (5.3% accuracy), likely because its single-layer transformer lacks sufficient capacity for this challenging neural decoding task. Classifying semantic categories while subjects read or imagine sentences is even more difficult. However, NeuroSketch achieves the best results, demonstrating robustness in challenging speech decoding scenarios. Specifically, compared to

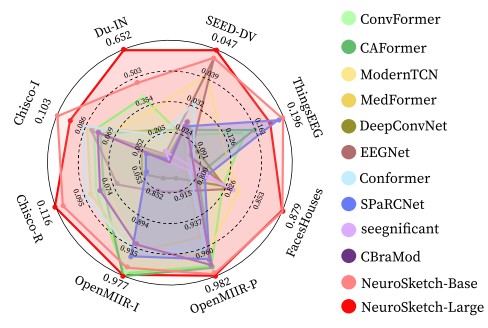

Figure 3: **Model performance comparison.**

the second-best baseline, Conformer, NeuroSketch-Large improves accuracy by 22.1% on Chisco-R and NeuroSketch-Base by 9.2% on Chisco-I.

Table 4: **Average classification accuracy** (mean ± std across three folds) for models on multiple neural decoding datasets. Higher throughput indicates faster inference. The best results are in **bold**. We use — to denote entries that are not applicable or not evaluated. Due to substantial differences between iEEG and EEG, we evaluate each pretrained model only on the modality it was pretrained on, the corresponding cross-modality cells are therefore marked with —. In addition, CBraMod uses a patch size of 200, whereas *ThingsEEG* has an input length of 100, so CBraMod cannot be run on *ThingsEEG* and is marked —.

| Models \ Datasets | Speech Decoding | | | Visual Decoding | | | Auditory Decoding | |
|---|---|---|---|---|---|---|---|---|
| | Du-IN | Chisco-R | Chisco-I | SEED-DV | Things-EEG | Faces-Houses | Open-MIIR-P | Open-MIIR-I |
| ConvFormer | .394±.005 | .086±.001 | .080±.004 | .024±.000 | .151±.003 | .820±.013 | .975±.005 | .976±.002 |
| CAFormer | .221±.003 | .081±.003 | .076±.002 | .025±.001 | .160±.009 | .800±.012 | .976±.006 | .977±.002 |
| ModernTCN | .339±.005 | .079±.003 | .074±.002 | .040±.001 | .123±.003 | .835±.006 | .964±.010 | .935±.002 |
| MedFormer | .212±.006 | .089±.003 | .085±.001 | .025±.000 | .045±.002 | .756±.011 | .948±.001 | .941±.001 |
| DeepConvNet | .053±.002 | .053±.004 | .052±.001 | .029±.001 | .091±.001 | .822±.005 | .896±.009 | .789±.005 |
| EEGNet | .060±.003 | .071±.002 | .082±.002 | .046±.002 | .070±.000 | .830±.006 | .915±.010 | .852±.006 |
| Conformer | .205±.003 | .095±.002 | .087±.002 | .033±.002 | .077±.002 | .762±.001 | .962±.013 | .930±.002 |
| SPaRCNet | .026±.001 | .005±.000 | .005±.000 | .026±.001 | .189±.001 | .805±.007 | .969±.007 | .948±.005 |
| seegnificant | .053±.002 | — | — | — | — | .831±.006 | — | — |
| CBraMod | — | .084±.002 | .077±.001 | .026±.002 | — | — | .975±.011 | .933±.008 |
| NeuroSketch-Base | .472±.006 | .111±.001 | **.103±.002** | .045±.002 | **.196±.002** | .879±.008 | .981±.010 | .970±.002 |
| NeuroSketch-Large | **.652±.002** | **.116±.001** | .095±.005 | **.047±.003** | .177±.001 | **.879±.007** | **.982±.009** | **.977±.002** |

**Visual decoding.** As shown in Table 4, for static image decoding on the FacesHouses dataset, NeuroSketch-Large outperforms the second-best performing baseline, ModernTCN, with a 5.2% improvement in accuracy. Furthermore, on the considerably more challenging ThingsEEG dataset, NeuroSketch-Base achieves competitive performance with the recent brain model, SPaRCNet. ConvFormer and CAFormer also demonstrate strong visual decoding performance on the ThingsEEG datasets. For the highly challenging video decoding task, where most baseline models perform near the chance level (2.5% accuracy), NeuroSketch-Large demonstrates a clear advantage with an accuracy of 4.7%. This result highlights its capacity to capture complex neural representations associated with dynamic visual stimuli.

**Auditory decoding.** The results in Table 4 show that NeuroSketch achieves state-of-the-art performance on both auditory decoding tasks. On the OpenMIIR-P dataset, the EEG foundation model, CBraMod attains the strongest baseline among brain-domain models (97.5% accuracy), highlighting the effectiveness of pretraining and its transferability to auditory decoding.

## 4 CONCLUSION AND FUTURE WORK

**Conclusion.** In this paper, we conduct an in-depth exploration of model architectures for neural decoding. By investigating basic architectures, latent space transformation, and computational methods optimization, we introduce NeuroSketch, an effective framework for neural decoding. Experimental results demonstrate that NeuroSketch achieves state-of-the-art performance across eight distinct neural decoding tasks. We hope this research will provide new insights for the neural decoding community and encourage further exploration of CNN-based architectures.

**Limitations and future work.** While NeuroSketch shows strong performance, its current design focuses on supervised learning. Given the favorable scaling behavior observed in Appendix F.3, we hypothesize that scaling both data and model capacity will yield further gains. Accordingly, we will explore large-scale pretraining strategies based on NeuroSketch in the future. Additionally, we plan to incorporate a broader range of neural decoding tasks to enhance the model's applicability across diverse neural decoding scenarios.

## 5 ETHICS STATEMENT

All datasets used in this study are publicly available. The corresponding dataset links are provided in Appendix C.1. Consequently, our work does not involve personally sensitive information and does not pose apparent ethical concerns.

## 6 REPRODUCIBILITY STATEMENT

We place great importance on ensuring the reproducibility of our work. To this end, details of the preprocessing of each dataset are provided in Appendix C.2, the detailed model implementation and configuration are described in Appendix D.2, and the experimental settings are outlined in Appendix E. The complete experimental results are available at `https://anonymous.4open.science/r/NeuroSketch_Results-BA5B`. We will release our code and scripts upon publication, thereby facilitating transparent and reproducible research for the community.

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

## A  THE USE OF LARGE LANGUAGE MODELS (LLMS)

In preparing this manuscript, we made limited use of a large language model (LLM) to assist with polishing the writing for grammar and clarity. In addition, the LLM was occasionally employed for small-scale code completion, such as generating boilerplate functions or suggesting minor syntax corrections. All core ideas, experimental designs, analyzes, and conclusions were completely conceived, implemented, and validated by the authors.

## B  RELATED WORK

**Neural decoding.** Decoding sensory experiences from neural signals is a central objective in neuroscience, spanning modalities such as auditory, speech, and visual processing, each with distinct decoding challenges and objectives. Auditory decoding can be divided into two distinct processes: auditory perception, which decodes brain activity evoked by external sounds (Stober et al., 2015; Sonawane et al., 2021), and auditory imagery, which identifies patterns associated with internally generated mental simulations of sound (Stober et al., 2015; De Borst et al., 2016). Speech decoding seeks to reconstruct vocal expressions or semantic content directly from neural signals, which can be

conducted at multiple linguistic levels, from syllables (Feng et al., 2023) to words (Zheng et al., 2024) and entire sentences (Zhang et al., 2024). Visual decoding focuses on reconstructing high-quality static images (Gaziv et al., 2022; Ahmadieh et al., 2024; Singh et al., 2023) or dynamic videos (Wen et al., 2018; Liu et al., 2024a) perceived by the human visual system from neural signals.

**Framework design.** Recent advances in neural architecture design have diverged into two directions: automated neural architecture search (NAS) and manual design driven by experience. Tan and Le (2019) utilize NAS to scale all dimensions of depth, width, and resolution through an effective compound coefficient, leading to a family of models known as EfficientNets. Radosavovic et al. (2020) further integrate NAS with manual design principles to explore the design space, resulting in the development of RegNet. Meanwhile, manual architectural innovations revitalize traditional components. Yu et al. (2022) reveals that instead of a specific token mixer(e.g., attention), the general architecture of Transformers, termed MetaFormer, is more essential for achieving high performance. Liu et al. (2022) modernize a standard CNN towards the design of Swin Transformer through macro- and micro-level optimizations, and ultimately propose a family of pure CNNs dubbed ConvNeXt.

## C  DETAILS OF DATASETS

### C.1  DATASET DESCRIPTION

In this study, we use six neural decoding datasets, each associated with specific experimental tasks. Brief descriptions of each dataset are provided below.

**Du-IN** (Zheng et al., 2024) focuses on decoding spoken Mandarin words. In each trial, a word is first displayed in white for 0.5 seconds, then turns green for 2 seconds to prompt articulation. Participants are instructed to read the word aloud during this cue period. Following this, the word disappears, and a fixation cross is presented. EEG data are recorded from 12 participants using 7-13 SEEG electrodes, comprising 72–158 channels. Each participant reads 61 predefined Chinese words, each word repeated 50 times, while both SEEG and audio signals are recorded simultaneously. In total, each participant contributes approximately 15 hours of data sampled at 2000 Hz, with around 3 hours corresponding to task-related activity. These task-related segments are divided into 3000 three-second trials and subsequently downsampled to 1000 Hz. The dataset is licensed under CC BY 4.0 and is available at `https://huggingface.co/datasets/liulab-repository/Du-IN`.

**Chisco** (Zhang et al., 2024) focuses on decoding silent reading and imagined speech. Data are collected from three healthy participants using a 125-channel EEG system at a sampling rate of 1000 Hz. The experiment consists of 45 blocks, each comprising 150 trials. Each trial includes a 5-second silent reading phase followed by a 3.3-second imagined speaking phase, with continuous EEG recording throughout. The stimuli include 6,681 commonly used Chinese sentences (ranging from 6 to 15 characters) drawn from 39 semantic categories. In each block, 10 trials are randomly selected for verbal recall. If a participant's recalled sentence differs from the original by more than four characters, the trial is marked as incorrect. If two or more errors occur within a block, the participant takes a rest and repeats the block. The dataset is licensed under CC0 and is available at `https://openneuro.org/datasets/ds005170/versions/1.1.2`.

**FacesHouses** (Miller, 2019) focuses on visual decoding of static images. ECoG signals are recorded from 14 epilepsy patients using subdural electrode strips implanted over the inferior temporal cortex. Participants view randomized grayscale images of faces and houses. The sampling rate is 1000 Hz. Each experiment includes three sessions, each presenting 50 face images and 50 house images for 400 ms, with a 400 ms blank interval between images. The dataset is licensed under CC BY-SA and is available at `https://purl.stanford.edu/zk881ps0522`.

**ThingsEEG** (Gifford et al., 2022) focuses on complex image decoding using a RSVP paradigm. EEG is recorded from 10 healthy participants using a 64-channel system at 1000 Hz. Stimuli consist of 16,740 image categories (16,540 for training and 200 for testing). Each training image is shown four times, while each test image is shown 80 times. Images are presented at a 200-ms stimulus onset asynchrony (SOA). A target detection task is included to maintain participant engagement. The decoding analysis is performed on the 200 test images. The dataset is licensed under CC BY 4.0 and is available at `https://osf.io/hd6zk/`.

**SEED-DV** (Liu et al., 2024a) focuses on decoding dynamic video stimuli. EEG is recorded from 20 participants at 1000 Hz while they watch video clips. The stimulus set consists of 1,400 two-second video clips spanning 40 semantic concepts, with 35 clips per concept. Each participant completed seven video blocks, with rest intervals between blocks. Each block contains all 40 concepts presented in a randomized order. Before the start of each block, participants are informed of the target concept and subsequently watch five video clips corresponding to that concept. The dataset is available after submitting an application at `https://bcmi.sjtu.edu.cn/ApplicationForm/apply_form/`.

**OpenMIIR** (Stober et al., 2015) focuses on music perception and imagination. EEG signals are recorded from 10 participants (8 with formal music training) using a 64-channel system at a sampling rate of 512 Hz. The experiment comprises two sessions, each consisting of five trials. In each trial, 12 musical stimuli were randomly presented under one of four experimental conditions: (1) perception with auditory cues, (2) imagination with cues, (3) imagination without cues, and (4) imagination without cues but with feedback. The stimuli encompass two time signatures (3/4 and 4/4) and vary in tempo from 104 to 212 BPM. Based on lyrical content, the stimuli are further classified into songs with lyrics, their corresponding instrumental versions, and purely instrumental pieces. The dataset is licensed under ODC-PDDL and is available at `https://hyper.ai/en/datasets/5591`.

## C.2 DATA PREPROCESSING

To ensure consistency and comparability, we follow the preprocessing procedures described in the original publications for each dataset.

For the **Du-IN** dataset, we followed the preprocessing setup described in the original work. For each subject, we selected 10 SEEG channels following the original configuration. The signals were downsampled to 1000 Hz, with each trial spanning 2.5 seconds. The classification targets corresponded to the 61 predefined Chinese words presented to the participants, with the objective of decoding spoken content directly from neural activity.

For the **Chisco** dataset, we examined two tasks: silent reading (denoted as Chisco-R) and imagined speaking (denoted as Chisco-I). Following the protocol of the original work, we removed three noisy channels and retained the remaining 122 valid EEG channels. All signals were downsampled to 500 Hz. The input duration was set to 5 seconds for Chisco-R and 3.3 seconds for Chisco-I, aligned with the respective task lengths. For the classification targets, both datasets were labeled with the 39 predefined semantic categories.

For the **FacesHouses** dataset, we segmented the continuous ECoG recordings into epochs based on stimulus markers, retained only face and house trials, and applied channel-wise z-score normalization to reduce variability across channels. The input data comprised between 31 and 102 channels per subject, sampled at 1000 Hz with a duration of 400 ms per trial. The classification labels were defined as a binary distinction between face and house.

For the **ThingsEEG** dataset, we used the preprocessed data released by the original authors and defined a visual stimulus classification task. For each subject, we loaded the test set from the original Things-EEG dataset, which contained images from 200 distinct categories, with each image presented 80 times. This resulted in a recorded signal of shape [200, 80, 17, 100], where 17 was the number of channels and 100 the number of time steps. These 17 channels were selected from the occipital and parietal cortex (O1, Oz, O2, PO7, PO3, POz, PO4, PO8, P7, P5, P3, P1, Pz, P2, P4, P6, P8) to focus on regions most relevant for visual processing. We assigned labels to the corresponding trials and then reshaped the data into [16000, 17, 100] (i.e., 200 categories × 80 trials), with the corresponding labels reshaped into [16000]. The data were then randomly shuffled and split into training, validation, and test sets. Specifically, we used 20% of the data as the test set, while the remaining 80% were divided into three folds; in each round, one fold was used for validation and the other two for training. The model input was the EEG segment with shape [17, 100], and the target was a label from one of the 200 image categories. We conducted this decoding task separately for each subject.

For the **SEED-DV** dataset, we adopted the first benchmark task from the original SEED-DV study: 40-class classification of fine-grained video concepts, to evaluate dynamic visual stimulus decoding. Following the preprocessing protocol described in the original work, we downsampled the EEG signals to 200 Hz, with each trial spanning 2 seconds and containing 62 channels. The classification

labels correspond to the 40 predefined video concepts. To standardize our evaluation pipeline, we merged data across all blocks for each subject, randomly shuffled the trials, and partitioned them into training, validation, and test sets.

For the **OpenMIIR** dataset, we focused on two tasks: perception(denoted as OpenMIIR-P) and imagination (denoted as OpenMIIR-I). Following the preprocessing protocol of the original work, we used raw EEG signals sampled at 512 Hz without further downsampling. The data were band-pass filtered between 0.5–30 Hz, and trials were segmented based on audio onset events, with epochs defined for each musical stimulus under the corresponding condition. For the perception task, we used condition 1, while for the imagination task, we aggregated data from conditions 2, 3, and 4. After segmentation, the continuous recordings for each stimulus were further divided into non-overlapping windows of length 600 samples across 64 channels. We then applied channel-wise z-score normalization on each window. The classification labels were defined as the 12 distinct musical stimuli.

# D    DETAILS OF MODELS

## D.1    MODELS IN THE ROADMAP EXPLORATION

Here, we provide details of the models used during the optimization process described in Section 2.

**Basic architecture study.** The model depth and embedding dimension during the basic architecture study are shown in Table 5. In both the CNN-1D and CNN-2D architectures, the convolution kernel size is set to 3. In CNN-1D, the number of channels increases from 1 to 64 in the stem layer, followed by 4 stages, each consisting of 5 layers. The number of channels in these stages increases progressively from 128 to 256, 512, and finally 1024. In CNN-2D, the channel configuration also starts from 1 to 64 in the stem layer. This is followed by 4 stages, each containing 4 layers, with the number of channels increasing from 96 to 192, 384, and 768, respectively. In the CNN-GRU architecture, the ratio of convolutional layers to GRU layers is set to 3:1. In the CNN-Transformer architecture, the initial ratio of convolutional layers to Transformer modules is set to 4:1. To fairly investigate how short- and long-term temporal information impacts model performance, we fix the total number of layers at 20 and systematically adjust the proportion of Transformer modules. To systematically evaluate the effectiveness of the proposed patching method, we vary only the data segmentation approach while keeping the embedding and subsequent self-attention-based feature extraction pipeline unchanged. All other hyperparameters are fixed to ensure that the observed performance differences can be attributed to the effectiveness of the patching method.

Table 5: **Model configuration for basic architecture study.**

| Model | # Params (M) | Number of Layers | Hidden Dimension |
|---|---|---|---|
| CNN-1D | 30 | 20 | 1024 |
| CNN-2D | 35 | 16 | 768 |
| GRU | 25 | 4 | 512 |
| Transformer | 20 | 8 | 512 |
| PatchTST | 20 | 8 | 512 |
| iTransformer | 20 | 8 | 512 |
| CNN-GRU | 34 | 16 | 512 |
| CNN-Transformer | 32 | 20 | 768 |
| CNN-Transformer v2 | 36 | 20 | 1024 |

**Macro optimization.** From a macro perspective, we optimize the transformation of the latent space during forward propagation. Specifically, we investigate two strategies for increasing the number of feature maps: the step approach and the leap approach. In the step approach, the number of feature maps is gradually increased: starting from 1 to 64 through the stem layers, then progressively to 96, 192, 384, and 768 across 4 stages, each containing 4 layers. In contrast, the leap approach increases the number of feature maps from 1 to 64 in the stem layer, and then directly increases it to 384 using an embedding layer. Subsequently, 16 layers are used to refine high-level features. We also investigate two strategies for decreasing the size of feature maps: the pyramid approach and the

pagoda approach. In the pyramid approach, a downsampling layer is placed in each of the last three stages, with a downsampling ratio of 2 at each stage. In contrast, the pagoda approach places three identical downsampling layers consecutively within the second stage.

**Micro optimization.** From a micro perspective, we optimize the computation method. Specifically, we investigate two variants of vanilla convolution: the group convolution and the separable convolution. For the group convolution, the number of groups is set to 4. For the separable convolution, the number of groups is set to 1, followed by a pointwise convolution with a kernel size of 1 to aggregate channel-wise information.

## D.2 NeuroSketch Implementation

Here, we provide a detailed description of the final NeuroSketch architecture to ensure transparency and reproducibility. Given an input tensor $\mathbf{X}$ of shape $\mathbb{R}^{B \times C \times L}$, where $B$ is the batch size, $C$ is the number of channels, and $L$ is the temporal length, we first reshape it to $\mathbb{R}^{B \times 1 \times 3C \times L//3}$ to form a 2D representation. The reshaping operation aligns with the subsequent convolutional kernel sizes and ensures that the temporal structure of the data remains intact in the first stem layer. The reshaped 2D representation is then passed through a stem stage consisting of four Conv2D–BatchNorm–ReLU blocks with kernel sizes [3, 3, 3, 3], paddings [1, 1, 1, 1], and strides [2, 1, 1, 2]. The input/output channels for the four blocks are $[1 \to 64]$, $[64 \to 256]$, $[256 \to 64]$, $[64 \to 96]$.

Following the stem stage, there are four feature extraction stages. The input/output channel dimensions for these stages are $[96 \to D_{\text{stage 1}}]$, $[D_{\text{stage 1}} \to D_{\text{stage 2}}]$, $[D_{\text{stage 2}} \to D_{\text{stage 3}}]$, $[D_{\text{stage 3}} \to D_{\text{stage 4}}]$. Each stage contains $d$ blocks with two key components:

1. **Patch Embedding:** In the first block of each stage, a 2D convolution with kernel size 3 and stride 1 maps the input channels to the output channels, followed by batch normalization. For the first three blocks of the second stage, we use a stride of 2 to downsample the input; in other cases, the layer reduces to an identity mapping when no change in resolution is needed.

2. **Convolution Module:** This component applies grouped $3 \times 3$ convolutions (with the number of groups equal to $G$) to efficiently capture local channel-wise dependencies. It is followed by batch normalization, ReLU activation, and a $1 \times 1$ convolution for feature fusion. The output is added back to the patch-embedded input via a residual connection.

After the four stage of feature extraction, we obtain a tensor of shape $\mathbb{R}^{B \times D_{\text{stage 4}} \times C' \times T'}$, where $C'$ and $T'$ denote the downsampled channel and temporal dimensions. We then apply GeM pooling to aggregate it into a representation $\mathbb{R}^{B \times D_{\text{stage 4}}}$, which is finally passed through a linear layer to produce the class probabilities. We implement two variants of the architecture: NeuroSketch-Base and NeuroSketch-Large. Their configurations are summarized in Table 6.

Table 6: **Detailed model configuration for NeuroSketch-Base and NeuroSketch-Large.**

|  | NeuroSketch-Base | NeuroSketch-Large |
|---|---|---|
| # Params (M) | 1.4 | 4.2 |
| $D_{\text{stage 1}}$ | 96 | 96 |
| $D_{\text{stage 2}}$ | 128 | 144 |
| $D_{\text{stage 3}}$ | 160 | 256 |
| $D_{\text{stage 4}}$ | 192 | 384 |
| $d$ | 2 | 3 |
| $G$ | 4 | 4 |

## D.3 Baseline Models

Here, we introduce the details of the baselines for performance evaluation in Section 3.

**ConvFormer** and **CAFormer** (Yu et al., 2023) are two variants of MetaFormer (Yu et al., 2022), employing different token mixers. ConvFormer uses depthwise separable convolution as its token

mixer. With this design, ConvFormer can be regarded as a pure CNN model that does not rely on channel or spatial attention mechanisms. CAFormer uses depthwise separable convolution as the token mixer in the first two stages of the model and self-attention in the last two stages. This design enables CAFormer to capture local features while better obtaining long-range dependencies.

**ModernTCN** (Luo and Wang, 2024) is a modernized purely convolutional architecture. It enhances the traditional TCN by incorporating depthwise convolutions and convolutional feed-forward networks (ConvFFNs) into the 1D CNN design. Additionally, it introduces time series–specific adaptations, such as patchified variable-independent embeddings, to improve suitability for time series tasks.

**MedFormer** (Wang et al., 2024) is a multi-granularity patching Transformer specifically designed for medical time series classification. It effectively captures the features of medical time series through cross-channel patching, multi-granularity embedding, and a two-stage multi-granularity self-attention mechanism.

**DeepConvNet** (Schirrmeister et al., 2017) is a deep CNN tailored for EEG decoding. It begins with a temporal convolution to capture frequency-specific patterns, followed by a spatial convolution across channels to model inter-channel dependencies. A stack of convolution–batch normalization–ELU–max pooling blocks with dropout then learns hierarchical spatio-temporal representations, and a final dense layer performs classification. This end-to-end design avoids handcrafted features and is effective across BCI tasks.

**EEGNet** (Lawhern et al., 2018) is a compact CNN tailored for EEG-based BCI. It decouples frequency and spatial filtering by using temporal convolutions as learnable band-pass filters followed by depthwise spatial convolutions across channels, and employs separable (pointwise) convolutions for efficient feature mixing. EEGNet attains strong performance across diverse BCI paradigms while using fewer than 0.5 million parameters, making it data-efficient and well suited to small EEG datasets.

**ConFormer** (Song et al., 2022) is a CNN-Transformer model for EEG signals. The convolution module learns low-level local features through one-dimensional temporal and spatial convolution layers, and the self-attention module processes the output of the convolution module to learn global temporal dependencies.

**SPaRCNet** (Jing et al., 2023) is a 1D CNN for seizure pattern recognition. Its structure features dense and transition blocks. Each dense block has four layers of two convolutional layers and two ELUs, and each transition block contains an ELU, a convolutional layer, and an average pooling layer.

**seegnificant** (Mentzelopoulos et al., 2024) is a brain foundation model for cross-subject neural decoding from SEEG. It tokenizes electrode-wise signals using convolutions, models long-term temporal dependencies with self-attention, and integrates electrode 3D spatial locations through positional encoding followed by cross-electrode attention. A unified backbone extracts global neural representations, while subject-specific task heads enable individualized decoding. Trained on multi-session SEEG data from 21 participants, seegnificant demonstrates effective behavioral response time decoding and supports few-shot transfer to new subjects, offering a path toward robust multi-subject generalization in SEEG analysis.

**CBraMod** (Wang et al., 2025b) is a brain foundation model for EEG decoding. The model first segments EEG signals into patches and randomly masks them. After patch encoding and asymmetric conditional position encoding, it learns spatio-temporal dependencies through the parallel spatial and temporal attention mechanisms of the cross-Transformer. Finally, it uses the reconstruction head to reconstruct the masked EEG patches, thereby learning the general representation of EEG signals.

# E EXPERIMENTAL SETTINGS

Here, we present the training strategies employed for the models described in Section 2 and Section 3.

### E.1 DATA SPLITTING

For Section 2, each subject's data is split into training, validation, and test sets with a 3:1:1 ratio. For Section 3, we allocate 20% of the data for testing, and the remaining 80% is used for a 3-fold cross-validation. Specifically, the remaining data are divided into three equal parts, with each iteration using one part as the validation set while the other two are combined for training, ensuring a comprehensive and reliable assessment of the model's effectiveness.

### E.2 DATA AUGMENTATION

To further enhance data diversity, we employ a strong data augmentation strategy comprising the following techniques:

**Random Shift.** We define a maximum shift range proportional to the input sequence length. For each training instance, a shift step is randomly sampled from this range. A positive value indicates a forward shift of the sequence, whereas a negative value corresponds to a backward shift. This approach enhances the model's robustness to uncertainty in stimulus onset times.

**Noise.** We generate noise from a standard normal distribution matching the shape of the original data. The noise is scaled by a predefined standard deviation and added to the input to produce noisy data. This method improves the model's resilience to signal perturbations.

**Channel Masking.** During training, a mask is applied to each channel with the specified probability, zeroing out the corresponding channel values. This technique reduces over-reliance on specific channels and promotes better integration of multi-channel information.

**Time Masking.** Similar to channel masking, we apply masks along the temporal dimension. This encourages the model to extract features robustly across various time segments and enhances generalization.

**Mixup.** A mixing coefficient $\lambda$ is sampled from a Beta distribution parameterized by a hyperparameter $\alpha = 0.4$. The original sample and a randomly chosen sample are linearly combined using $\lambda$, and their corresponding labels are mixed accordingly. This augmentation method exposes the model to a broader range of data combinations, thereby improving generalization.

### E.3 HYPERPARAMETERS

Table 7: **Training hyperparameters.**

| Hyperparameters | Settings |
|---|---|
| Epoch | 100(Chisco), 500(others) |
| Batch size | 64 |
| Seed | 42 |
| Optimizer | AdamW |
| Learning rate | 1e-3 |
| Weight decay | 5e-2 |
| Scheduler | Cosine |
| Warmup ratio | 0.1 |
| Early stop ratio | 0.2 |
| Random shift probability | 0.5 |
| Random shift ratio | 0.2 |
| Add noise probability | 0.1 |
| Channel masking probability | 0.5 |
| Time masking probability | 0.5 |
| Mixup probability | 0.5 |

Detailed hyperparameters are shown in Table 7. Across multiple datasets, we observed that decoding performance typically improves with an increased number of training epochs. Based on this observa-

tion, we set the number of training epochs to 500 in most experiments to fully optimize the model performance. However, for the *Chisco* dataset, the accuracy reaches a plateau at around 80 epochs and shows no further improvement. Therefore, we limit training on *Chisco* to 100 epochs.

### E.4 COMPUTE RESOURCES

We utilized an AMD EPYC 7663 56-core processor and eight NVIDIA A100 GPUs, each with 80 GB of memory. Section 2 outlines 21 distinct models, each evaluated on 84 experiments that together span the entire set of subjects across eight decoding tasks. In total, this results in 1,764 experiments. Each experiment is executed on an A100 GPU and requires, on average, one hour of training time. Section 3 evaluates 12 distinct models, each trained with three cross-validation folds, resulting in a total of 3,024 experiments. On average, each experiment requires approximately 50 minutes of training time.

## F EXPERIMENTAL ANALYSIS

### F.1 NEUROPHYSIOLOGICAL INTERPRETABILITY

To further investigate the spatiotemporal features captured by NeuroSketch, we conducted a case study on the Du-IN dataset. In this analysis, we trained NeuroSketch-Large on all 115 channels from subject 02 and 90 channels from subject 11, and applied the Score-CAM (Wang et al., 2020) method to visualize the model's decision-making process across both spatial and temporal dimensions.

Score-CAM generates class-activation maps by computing the gradient-free importance of each feature map and projecting it back to the input space, thereby highlighting which spatiotemporal regions most strongly influence the predicted class. For each sample, we obtained a saliency map and subsequently averaged these maps across all samples from subject 02 and subject 11, respectively, to produce subject-wise saliency maps. These aggregated maps, as illustrated in Figure 4, were then used to identify regions of interest (ROIs) within both the spatial and temporal domains.

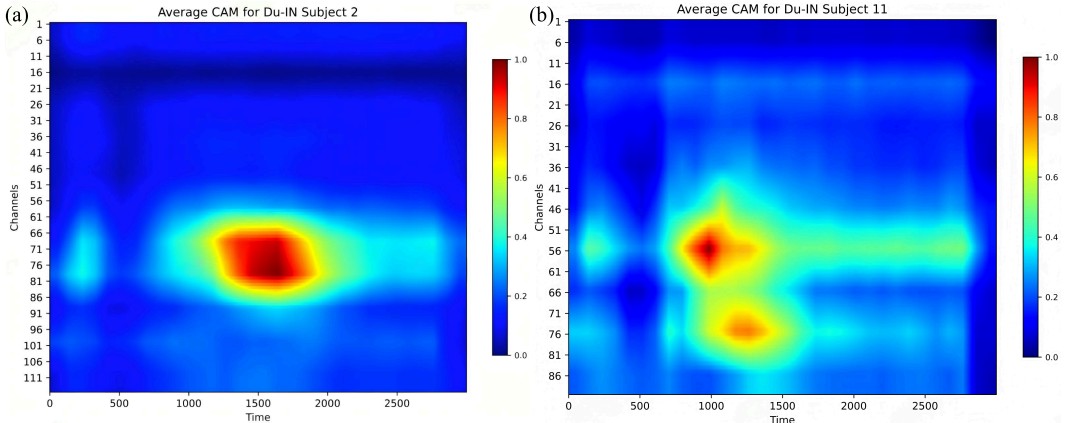

Figure 4: Score-CAM visualization of all channels for Subjects 02 and 11 in the Du-IN dataset generated by NeuroSketch-Large.

For spatial analysis, we averaged each subject-wise saliency map across the temporal dimension to obtain a saliency score for every channel. Notably, for subject 11, the channels previously reported as significant in the original study (Zheng et al., 2024) were 52,55,56,57,65,74,75,76,77,78. Except for channel 65, all nine of these channels appeared within our top-20 most salient channels identified by NeuroSketch. Similarly, for subject 02, the significant channels reported in Zheng et al. (2024) were 72,73,74,75,76,77,100,109,110,111, among which channels 72–77 were also ranked within our top-20. Beyond these specific electrodes, the remaining highly salient channels detected by our model were found to be spatially clustered around these previously reported regions, indicating that NeuroSketch captures spatial activity patterns that are consistent with experimentally validated cortical regions.

More importantly, the top-contributing electrodes identified by NeuroSketch are located within or near the ventral sensorimotor cortex (vSMC) and the bilateral superior temporal gyrus (STG)—two cortical regions well established as the core network for speech motor control (Bouchard et al., 2013; Hickok and Poeppel, 2007; Chartier et al., 2018). The vSMC is primarily responsible for the motor coordination of articulatory movements, while the STG supports auditory feedback processing and vocal self-monitoring during speech production. The focus of the model on these regions highlights its ability to capture biologically interpretable and functionally grounded spatial activation patterns that align with established speech-related cortical circuits, further validating the neurophysiological relevance of our saliency findings.

For temporal analysis, the model exhibits a pronounced activation hotspot concentrated around the mid-trial period, approximately between 800-1600 ms for subject 11 and 1200-1800 ms for subject 02, followed by a gradual decrease in saliency toward both the early and late segments. This pattern indicates that NeuroSketch focuses on short-range dependencies rather than allocating attention uniformly over time, which aligns with the transient temporal dynamics of neural signals. Collectively, these results confirm the model's ability to extract interpretable and neurophysiologically grounded spatiotemporal features.

## F.2 Representation Analysis of CNN-2D and Transformer-Based Architectures

Given the observed performance gap between CNN-based and Transformer-based architectures from an empirical perspective in Section 2.2 and Section 3.2, we further analyzed their representations to uncover the underlying reasons behind this gap.

Specifically, we examined the model representations from a rank-based perspective, following the framework proposed by Yu et al. (2025). The rank of a feature representation reflects its expressive capacity: a higher rank indicates richer and more diverse features, while a lower rank implies excessive compression and loss of independent information. In a well-organized hierarchy, the rank is expected to remain high or increase with depth, as deeper layers capture more abstract yet independent features. In our evaluation, since the CNN-2D-based model and the Transformer-based model achieved the overall best and worst performance, respectively, we conducted an analysis on NeuroSketch-Large (CNN-2D-based) and MedFormer (Transformer-based) to examine how the rank of the feature representations produced by each layer evolves across network depth.

For any layer of a given model, let the output representation of a sample be denoted as $\mathbf{O} \in \mathbb{R}^{s \times d}$, where $s$ represents the sequence length and $d$ the feature dimension. We performed singular value decomposition (SVD) on $\mathbf{O}$ as $\mathbf{O} = \mathbf{U}\Sigma\mathbf{V}^T$, where $\mathbf{U}$ and $\mathbf{V}$ are orthogonal matrices, and $\Sigma$ is a diagonal matrix whose diagonal entries $\sigma_1 \geq \sigma_2 \geq \cdots \geq \sigma_{\min\{s,d\}}$ are the singular values. Although the algebraic rank—i.e., the number of nonzero singular values—serves as a strict measure of rank, real-world data are often noisy. Therefore, we adopt a more practical numerical rank. For a tolerance $\varepsilon > 0$, the $\varepsilon$-rank of $\mathbf{O}$ is defined as the number of singular values that are significant relative to the largest one:

$$\varepsilon\text{-rank}\,(\mathbf{O}) = \left| \left\{ i \Big| \frac{\sigma_i\,(\mathbf{O})}{\sigma_1\,(\mathbf{O})} > \varepsilon \right\} \right|, \tag{1}$$

Since the maximum rank (defined as $\min(s, d)$) varies across layers depending on $s$ and $d$, we further compute the $\varepsilon$-rank ratio, defined as the $\varepsilon$-rank relative to the maximum rank. This metric allows for a more intuitive comparison of how the effective rank evolves across layers.

$$\varepsilon\text{-rank ratio} = \frac{\varepsilon\text{-rank}\,(\mathbf{O})}{\min\,(s, d)}, \tag{2}$$

For MedFormer, we extracted the $\varepsilon$-rank ratios of the embedding layer and each of its six encoder layers under three different patch lengths (5,10 and 20) to examine how patch length affects representational rank. For NeuroSketch-Large, we analyzed the outputs of its three stem layers and all layers across the four feature-extraction stages, transforming the height–width dimensions of each feature map into a single sequence length for a consistent rank analysis.

As shown in Figure 5, from the two models' evolution of the $\varepsilon$-rank ratios, we can observe that (1) For multi-channel EEG signals, the embeddings are not as low-rank as those typically observed in conventional time-series (Liang et al., 2025; Yu et al., 2025). Specifically, for both MedFormer and

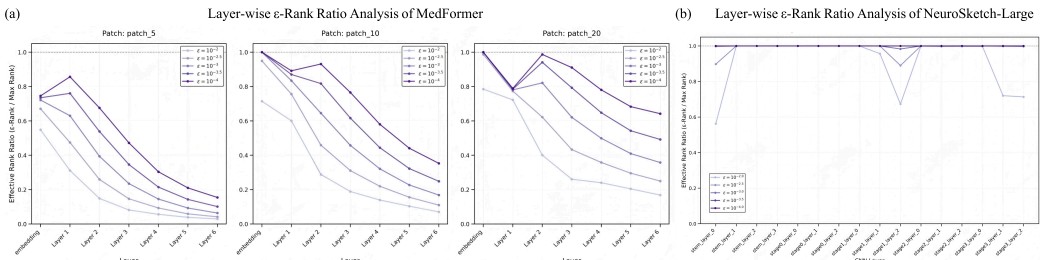

Figure 5: Layer-wise $\varepsilon$-Rank Ratio Analysis of MedFormer and NeuroSketch-Large.

NeuroSketch-Large, when $\varepsilon \leq 10^{-2.5}$, the $\varepsilon$-rank of the embedding layer (for MedFormer) and the first stem layer (for NeuroSketch-Large) remains above 60% of the maximum possible rank, indicating that both models retain high-dimensional representations rather than collapsing into a low-rank subspace. Moreover, increasing MedFormer's patch length from 5 to 20 raises the maximum $\varepsilon$-rank ratio rises accordingly, suggesting that longer temporal patches promote richer representations. We attribute these two phenomena in the information compression ratio during the initial embedding. For example, in MedFormer, the embedding stage applies a 2D convolution kernel of shape $[C, P]$, where $C$ is the number of input channels, and $P$ is the patch length. The convolution output is projected into an embedding space of dimension $D$. Consequently, the effective compression ratio is approximately $\frac{C \times P}{D}$. Given that brain signals typically involve a large number of channels (often 64 or more), this ratio is much higher than in univariate time-series data. Increasing $P$ further amplifies the compression, producing embeddings with higher numerical rank which is consistent with our observations.

(2) For both NeuroSketch and MedFormer, we observe distinct trends in how the $\varepsilon$-rank ratio evolves across layers. As the network depth increases, MedFormer tends to progressively reduce the rank of its feature representations, indicating excessive compression of the input information into a lower-dimensional subspace. In contrast, NeuroSketch exhibits the opposite trend: the rank of its representations gradually increases, approaching full rank in deeper layers. These contrasting trends imply that the two architectures differ fundamentally in how they organize their representation spaces. In neural decoding tasks, compared with the transformer architecture, the CNN-2D architecture is able to compress more robust and information-rich representations.

### F.3 SCALING BEHAVIOR

In this section, we analyze the scaling behavior of NeuroSketch. As shown in Table 4, NeuroSketch-Large achieves a clear performance improvement over NeuroSketch-Base on the Du-IN dataset, whereas the two models perform comparably on the other datasets. To further investigate this phenomenon, we examine subject-level performance on the Du-IN dataset, with the results summarized in Table 8. We observe that NeuroSketch-Large outperforms NeuroSketch-Base across all subjects, although the extent of improvement varies. To intuitively examine the relationship between the degree of improvement and the difficulty of the samples, we plot a scatter diagram of accuracy versus improvement (Figure 6). The scatter plot reveals an overall negative correlation between accuracy and improvement; in other words, the more difficult the samples are to classify, the more pronounced the effect of scaling becomes. Notably, for subjects with NeuroSketch-Base accuracy below 5%, NeuroSketch-Large yields relative improvements of over 400%. This indicates that scaling up NeuroSketch effectively addresses more challenging tasks.

### F.4 HYPERPARAMETER ANALYSIS

Since the architecture of NeuroSketch is based on the CNN-2D backbone, convolutional operations play a central role in its performance. To better understand their impact, we conducted a hyperparameter analysis focusing on two key factors: the kernel size and the number of groups in group convolutions.

**Kernel size.** The size of the convolutional kernel directly affects the receptive field and therefore controls the extent of local temporal–spatial dependencies that the model can capture. In NeuroSketch-

Table 8: **Scaling behavior of NeuroSketch on the Du-IN dataset.** We report the mean accuracy together with the standard variance, computed over three distinct cross-validation folds. Additionally, we indicate the relative improvement in accuracy of NeuroSketch-Large compared to NeuroSketch-Base.

| Subject | NeuroSketch-Base | NeuroSketch-Large | Relative Improvement |
|---|---|---|---|
| 1 | 0.626 ±0.018 | 0.791 ±0.027 | 26.37% |
| 2 | 0.761 ±0.025 | 0.798 ±0.007 | 4.95% |
| 3 | 0.081 ±0.025 | 0.508 ±0.016 | 529.34% |
| 4 | 0.422 ±0.052 | 0.765 ±0.033 | 81.36% |
| 5 | 0.774 ±0.012 | 0.882 ±0.013 | 13.90% |
| 6 | 0.209 ±0.021 | 0.465 ±0.016 | 122.49% |
| 7 | 0.409 ±0.019 | 0.561 ±0.021 | 37.28% |
| 8 | 0.525 ±0.012 | 0.630 ±0.021 | 20.00% |
| 9 | 0.540 ±0.030 | 0.817 ±0.005 | 51.36% |
| 10 | 0.034 ±0.008 | 0.183 ±0.003 | 433.01% |
| 11 | 0.721 ±0.013 | 0.758 ±0.011 | 5.09% |
| 12 | 0.569 ±0.050 | 0.668 ±0.010 | 17.39% |

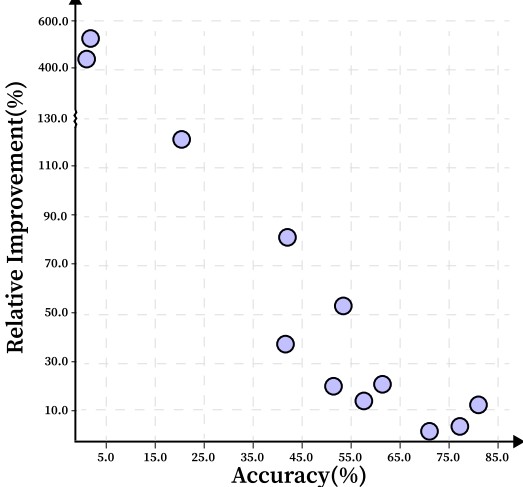

Figure 6: Scatter plot of the scaling behavior of NeuroSketch on the Du-IN dataset. The $x$ axis denotes the accuracy of NeuroSketch-Base for each subject and the $y$ axis represents the relative improvement achieved by NeuroSketch-Large.

Large, we adopt a kernel size of 3 as the default setting, since smaller kernels are generally more effective in capturing fine-grained dynamics while keeping the parameter overhead low. To further examine the effect of kernel size, we additionally evaluate larger kernels of 5 and 7 on the Du-IN dataset. The results are summarized in Table 9.

Table 9: **Results of different kernel sizes of NeuroSketch on the Du-IN dataset.** Results are reported as mean and standard deviation, computed across three distinct cross-validation folds.

| Kernel Size | Accuracy | Precision | F1 Score |
|---|---|---|---|
| 3 | $0.651 \pm 0.005$ | $0.667 \pm 0.003$ | $0.647 \pm 0.003$ |
| 5 | $0.557 \pm 0.005$ | $0.577 \pm 0.006$ | $0.550 \pm 0.005$ |
| 7 | $0.427 \pm 0.002$ | $0.456 \pm 0.002$ | $0.420 \pm 0.003$ |

These results indicate that larger kernel sizes lead to lower decoding accuracy, which aligns with our claim in Section 1 that neural decoding signals exhibit transient temporal dynamics. Larger kernels tend to capture longer-range temporal features, which may dilute short-term patterns that are critical for accurate decoding in this context.

**Number of groups.** The number of groups in group convolution determines how the feature channels are partitioned and processed, which in turn affects both the computational cost and the representational capacity. Increasing the number of groups reduces the computational cost, since fewer channels are convolved together within each group. In NeuroSketch-Large, we set the default group number to 4. To further examine the effect of this parameter, we additionally evaluate group numbers of 2, 8, and 16 on the Du-IN dataset. The results are summarized in Table 10.

Table 10: **Results of NeuroSketch under different number of groups in the group convolution on the Du-IN dataset.** Results are reported as mean and standard deviation, computed across three distinct cross-validation folds.

| #Group | Accuracy | Precision | F1 Score |
|---|---|---|---|
| 2 | $0.651 \pm 0.005$ | $0.667 \pm 0.003$ | $0.647 \pm 0.003$ |
| 4 | $0.652 \pm 0.002$ | $0.666 \pm 0.003$ | $0.648 \pm 0.002$ |
| 8 | $0.642 \pm 0.002$ | $0.658 \pm 0.003$ | $0.638 \pm 0.002$ |
| 16 | $0.642 \pm 0.001$ | $0.656 \pm 0.003$ | $0.637 \pm 0.002$ |

The results indicate that setting the group number to 4 provides the most favorable configuration. Compared with using 2 groups, this setting achieves competitive decoding performance while reducing the computational cost. At the same time, it clearly outperforms larger group numbers of 8 and 16 in terms of decoding accuracy, highlighting that 4 groups strike an effective balance between efficiency and representational capacity.

