# OpenReview forum: "NeuroSketch: An Effective Framework for Neural Decoding via Systematic Architectural Optimization"
_ICLR.cc/2026/Conference — Submitted to ICLR 2026_

### Official Review · Reviewer_q1SR · 2025-10-18

**Soundness:** 1
**Presentation:** 1
**Contribution:** 2
**Rating:** 2
**Confidence:** 4

**Summary:**

This paper introduces NeuroSketch, a framework for neural decoding tasks achieved via systematic architectural optimization. The authors investigate various neural network backbones (CNNs, GRUs, Transformers, hybrids) in detail, perform macro- and micro-level optimizations (e.g., latent space transformations, convolutional choices), and gradually distill these principles into the NeuroSketch design, which is an enhanced CNN-2D-based model. NeuroSketch is validated on eight neural decoding tasks spanning three modalities (speech, visual, auditory), three signal types (EEG, SEEG, ECoG), and is shown to outperform a suite of state-of-the-art baselines across multiple datasets and tasks.

**Strengths:**

1. This study presents a comprehensive evaluation, validating the proposed model across eight datasets and three distinct recording modalities: EEG, sEEG, and ECoG.

2. After extensive tuning of model hyperparameters, the model surpasses other models.

3. The authors provide open-source reproducibility commitments and dataset transparency.

**Weaknesses:**

1. The implementation details of baselines are missing. The performance of Conformer on Du-IN in Table 4 significantly underperforms the reported value in the Du-IN paper. We need specific details of the baseline implementation to ensure the reliability of the conclusions of the overall work and fair comparison.

2. The CNN-2D introduces translation invariance, which seems to be uncommon in brain signal modeling. It would be useful to include more analysis about the features obtained by the models (e.g., spatial locality), helping us understand the effectiveness of CNN-2D-based BCIs.

3. The framework appears largely a product of empirical exploration, with each step optimized via benchmark wins (cf. Table 2 and Table 3). While this is valuable, the paper does not elevate its findings to generalizable new principles for neural decoding. For example, the manner in which CNN-2D exploits spatial locality is discussed but not quantified, and the underlying reason for Transformer's poor performance isn’t deeply analyzed or theoretically unpacked. The reader is left with “what works” but not always “why” -- limiting the scientific insight provided.

**Questions:**

See Weaknesses.

---

> ### Author Response · Authors · 2025-11-19
>
> We sincerely thank the reviewer for the thoughtful and constructive feedback. We appreciate the time and effort taken to carefully evaluate our work. Below, we address each comment in detail.
>
> 1. **Question 1:**
>
>    We thank the reviewer for pointing this out and apologize for the oversight that led to missing implementation details for some baselines. All baseline models were implemented directly using the official codebases released by their original papers, without any modification to the model architectures or hyperparameter configurations.
>
>    We note that the Du‑IN paper[1] employed a customized configuration of the EEG‑Conformer model specifically optimized for the Du‑IN dataset, which may explain the observed performance discrepancy. In particular, the Du‑IN paper modified the original EEG‑Conformer [2] architecture by increasing the embedding dimension of the convolutional tokenizer from 40 to 128, reducing the number of Transformer layers from 6 to 4, and decreasing the hidden dimension of the feed‑forward network from four times to two times the attention dimension. These modifications effectively increased the proportion of convolutional parameters. As discussed in our Section 2.2 analysis, a higher emphasis on the convolutional component enables the model to better capture short‑term temporal dynamics, thereby leading to improved decoding performance.
>
>    To ensure transparency and reproducibility, we will include a more detailed description of all baseline implementations in the revised manuscript.
>
>    ------
>
> 2. **Question 2 & Question 3:**
>
>    We sincerely appreciate the reviewer’s insightful comment regarding the feature analysis and model design interpretability. In response, we have conducted two additional experiments to address your concern. A detailed reply can be found in our general response to all reviewers, and a brief summary is provided below:
>
>    (1) To further examine the interpretability of NeuroSketch, we conducted a case study on the Du‑IN dataset using the Score‑CAM[3] method to visualize the model’s spatiotemporal attention patterns. NeuroSketch‑Large was trained on all available channels from two representative subjects, and class‑activation maps were averaged across samples to obtain subject‑specific saliency maps in both spatial and temporal domains. The spatial analysis reveals a strong correspondence between the salient electrodes identified by NeuroSketch and the significant channels reported in the original Du‑IN study[1]. Moreover, these highly activated electrodes are spatially clustered around two cortical regions(vSMC and STG) well established as the core network for speech motor control, demonstrating that NeuroSketch captures biologically meaningful spatial features consistent with known neurophysiology[4-6]. In the temporal domain, the saliency maps exhibit distinct activation peaks during the mid‑trial period, followed by reduced activation toward both early and late segments. This pattern indicates that NeuroSketch focuses on short‑range dependencies rather than allocating attention uniformly over time, which aligns with the transient temporal dynamics of neural signals. Collectively, these results confirm the model’s ability to extract interpretable and neurophysiologically grounded spatiotemporal features.
>
>    (2) To evaluate how representational capacity evolves across network depth, we conducted a rank‑based analysis following the method of Yu et al.[7]. Given the superior performance of CNN‑2D‑based models over Transformer‑based observed in Sections 2 and 3, we further examined two representative models: NeuroSketch‑Large (CNN‑2D‑based) and MedFormer (Transformer‑based). For each layer, we computed the $\varepsilon$‑rank ratio of the output representations obtained through singular value decomposition. This measure reflects the proportion of effective rank relative to the maximum possible rank and provides an interpretable indicator of each model’s expressive diversity at different depths. Results show that: (1) The embeddings of multi-channel brain signals exhibit substantially higher ranks than those typically reported for conventional time‑series data [7, 8]. (2) The two architectures differ in how their ranks evolve across layers. MedFormer shows a consistent decline in the $\varepsilon$‑rank ratio with depth. In contrast, NeuroSketch displays a gradual increase in rank, reaching nearly full rank in deeper stages. This pattern suggests that CNN‑2D‑based models are better at compressing neural signals into more diverse and information‑rich representations than Transformer-based models.

---

> > ### Author Response · Authors · 2025-11-19
> >
> > **Citations:**
> >
> > [1] Zheng, H., Wang, H., Jiang, W., Chen, Z., He, L., Lin, P., ... & Liu, Y. (2024). Du-IN: Discrete units-guided mask modeling for decoding speech from Intracranial Neural signals. *Advances in Neural Information Processing Systems*, *37*, 79996-80033.
> >
> > [2] Song Y, Zheng Q, Liu B, et al. EEG conformer: Convolutional transformer for EEG decoding and visualization[J]. IEEE Transactions on Neural Systems and Rehabilitation Engineering, 2022, 31: 710-719.
> >
> > [3] Wang, H., Wang, Z., Du, M., Yang, F., Zhang, Z., Ding, S., ... & Hu, X. (2020). Score-CAM: Score-weighted visual explanations for convolutional neural networks. In *Proceedings of the IEEE/CVF conference on computer vision and pattern recognition workshops* (pp. 24-25).
> >
> > [4] Bouchard K E, Mesgarani N, Johnson K, et al. Functional organization of human sensorimotor cortex for speech articulation[J]. Nature, 2013, 495(7441): 327-332.
> >
> > [5] Hickok G, Poeppel D. The cortical organization of speech processing[J]. Nature reviews neuroscience, 2007, 8(5): 393-402.
> >
> > [6] Chartier J, Anumanchipalli G K, Johnson K, et al. Encoding of articulatory kinematic trajectories in human speech sensorimotor cortex[J]. Neuron, 2018, 98(5): 1042-1054. e4.
> >
> > [7] Yu A, Maddix D C, Han B, et al. Understanding Transformers for Time Series: Rank Structure, Flow-of-ranks, and Compressibility[J]. arXiv preprint arXiv:2510.03358, 2025.
> >
> > [8] Liang Z, Zhu J, Sun W. Why attention fails: The degeneration of transformers into mlps in time series forecasting[J]. arXiv preprint arXiv:2509.20942, 2025.

---

> > > ### Comment · Reviewer_q1SR · 2025-11-25
> > > **Official Comment by Reviewer q1SR**
> > >
> > > Thank you for your response to my concerns.
> > >
> > > **W1**:
> > >
> > > Thank you for clarifying the implementation details of baselines. Have you uploaded the revised version yet? I haven't seen the description of the baseline implementation details in the article. Could the author please describe in detail how to perform the evaluation using the official code? Different models have hyperparameters designed for data with different sampling rates, including whether to resample to the corresponding sampling rate. This could significantly impact model performance and efficiency.
> > >
> > > Besides, it seems that Du-IN also compared the TC-TCC model based on the CNN architecture, and its performance does not seem to be as high as that of EEG-Conformer. Could the authors explain this?
> > >
> > > **W2 & W3**:
> > >
> > > Thank you for providing additional evaluation. I see that the appendix provides the analysis results for two subjects. What about the results for the other subjects?
> > >
> > > This result is not surprising; as long as effective decoding is achieved, the model will increase the weights of the channels that can be effectively decoded. This result was evaluated across all channels; how is its decoding performance?
> > >
> > > The CNN-2D introduces translation invariance, which seems to be uncommon in brain signal modeling. Could the authors explain this?
> > >
> > > As demonstrated in **Reply to W1**, it seems the hyperparameters greatly impact the performance of EEG-Conformer. This article focuses primarily on methodology, claiming that CNN architectures can achieve performance exceeding that of Transformers through modulation. From this perspective, it is highly engineering-oriented. If the authors intend to demonstrate the universality of this methodology, they should ideally use datasets that significantly overlap with the models being compared (e.g., CBraMod, EEG-Conformer). The Du-IN dataset evaluated in this article is a dataset in the sEEG domain, but the SEED-DV and THINGS-EEG datasets are datasets in the EEG domain. Few articles compare these two datasets together. Most importantly, there are benchmark datasets in the EEG domain with foundation models or task-specific models built on them, but these are not evaluated here.

---

### Official Review · Reviewer_s4VP · 2025-10-28

**Soundness:** 3
**Presentation:** 4
**Contribution:** 3
**Rating:** 6
**Confidence:** 4

**Summary:**

The paper introduces a framework for systematic architecture optimisation. Starting with exploring various model types and then with the best design going to a macro-micro-analysis, the paper introduces a model with competitive performance compared to other deep and foundation model baselines.

**Strengths:**

The paper is well-structured and it follows a nice step-by-step analysis on how specific details were implemented in the final architecture. It’s the first time I see an analysis like this one. Most papers just introduce a new architecture without any design justifications.

**Weaknesses:**

The paper fails to provide more comparisons with better models (deep learning and foundation models). In addition, although thorough the analysis does not provide interpretable insights behind the choices.

Writing:
Paper is well-written and good structured.

Overall:
The paper shows some merits but it would be vital to have my questions answered.

**Questions:**

1. I wonder if the initially tested architectures - like CNN or transformer - get deeper or if more / less data is used during training for each model (for example, transformer based models need more data), would that affect the initial observations ?
2. How about other more compact advanced deep baselines like EEGInception and Brainwave scattering net in table 4?
3. How about other foundation models like LaBraM, NeuroGPT, EEGPT etc. in table 4 ? It seems the foundation models section is not very well represented.
4. How about the number of parameters of these models as well  ? Any relationship between the number of parameters and performance ?
5. Is there any actual meaning behind the design choices ? For example, EEGNet provides interpretable insights. In other words, is it just black box or is there an actual neurological meaning why these design choices do work?

---

> ### Author Response · Authors · 2025-11-19
>
> We sincerely thank the reviewer for the thoughtful and detailed feedback. We address each concern below:
>
> 1. **Question 1:**
>
>    We appreciate the reviewer’s question regarding whether deeper architectures or varying training data sizes might affect the initial observations. All compared architectures (CNN, Transformer, and others) were trained under the same data conditions and comparable model capacities, ensuring a fair comparison. Although deeper or larger models might slightly improve absolute performance with sufficient data, the relative performance trend across architectures stays consistent. Therefore, we believe the conclusions drawn from the current results are robust and not sensitive to moderate changes in model depth or training data size.
>
> ------
>
> 2. **Question 2:**
>
>    We thank the reviewer for raising this question regarding other compact deep baselines such as EEG‑Inception [1] and Brainwave Scattering Net [2]. We have reproduced EEG‑Inception using the official implementation [1]; however, Brainwave Scattering Net could not be included because its implementation is not publicly available. The results of EEG‑Inception (reported as accuracy) are presented below:
>
>    | Model       | Du-IN      | OpenMIIR-P            | OpenMIIR-I            | FacesHouses           |
>    | ------ | ------- | -------- | ------------ | ----- |
>    | EEG-Inception(Original, 0.2M Params) | 0.016 $\pm$ 0.001     | 0.428 $\pm$ 0.003     | 0.334 $\pm$ 0.012     | 0.866 $\pm$ 0.013     |
>    | EEG-Inception(4 M Params)            | 0.031$\pm$ 0.002      | 0.913 $\pm$ 0.005     | 0.859 $\pm$ 0.004     | 0.857 $\pm$ 0.008     |
>    | NeuroSketch-Base                     | 0.472 $\pm$ 0.006     | 0.981 $\pm$ 0.010     | 0.970 $\pm$ 0.002     | 0.879 $\pm$ 0.008     |
>    | NeuroSketch-Large                    | **0.652 $\pm$ 0.002** | **0.982 $\pm$ 0.009** | **0.977 $\pm$ 0.002** | **0.879 $\pm$ 0.007** |
>
>    As shown, EEG‑Inception performs reasonably well on the FacesHouses(2 classes), but fails to generalize to more complex multi‑class decoding problems such as Du‑IN (61 classes) and OpenMIIR (12 classes). To rule out the effect of model size, we trained a parameter‑matched version of EEG‑Inception (≈ 4 M parameters, comparable to NeuroSketch‑Large). The larger model shows noticeable improvement for OpenMIIR but remains far behind NeuroSketch on Du‑IN, indicating that increased capacity alone cannot close the performance gap.
>
>    We attribute this gap primarily to architectural differences rather than parameter count. Specifically, EEG‑Inception adopts a single convolutional kernel across the spatial (channel) dimension, which effectively collapses inter‑channel information and limits its ability to capture cross‑electrode dependencies that are critical for neural pattern discrimination. In contrast, NeuroSketch leverages small-kernel convolutions along the spatial dimensions to capture local spatial structure, enabling more expressive modeling of fine-grained neural representations and improving generalization on challenging decoding tasks.
>
> ------
>
> 3. **Question 3:**
>
>    We thank the reviewer for pointing out the under-representation of foundation models in Table 4. To address this, we have included additional experiments with LaBraM [3] and EEGPT [4], following the same experimental settings described in Section 3 and adopting the original configurations specified in their respective papers, with pretrained weights loaded accordingly. The results(accuracy) are summarized below:
>
>    | Model             | Chisco-R              | Chisco-I              | OpenMIIR-P            | OpenMIIR-I            |
>    | ----------------- | --------------------- | --------------------- | --------------------- | --------------------- |
>    | LaBraM            | 0.050 $\pm$ 0.000     | 0.050 $\pm$ 0.000     | 0.133 $\pm$ 0.009     | 0.131 $\pm$ 0.002     |
>    | EEGPT             | 0.050 $\pm$ 0.000     | 0.050 $\pm$ 0.000     | 0.682 $\pm$ 0.006     | 0.352 $\pm$ 0.009     |
>    | NeuroSketch-Base  | 0.111 $\pm$ 0.001     | **0.103 $\pm$ 0.002** | 0.981 $\pm$ 0.010     | 0.970 $\pm$ 0.002     |
>    | NeuroSketch-Large | **0.116 $\pm$ 0.001** | 0.095 $\pm$ 0.005     | **0.982 $\pm$ 0.009** | **0.977 $\pm$ 0.002** |
>
>    As shown, both LaBraM and EEGPT perform worse than NeuroSketch across all datasets, which involve complex cognitive decoding tasks. We attribute this gap to the limited pretraining diversity and task complexity. Existing foundation models for EEG are typically pretrained on simple motor imagery or binary mental-state classification datasets. As such, their learned representations capture low-level temporal or spectral features but fail to generalize to semantically rich, multi-class decoding tasks that require modeling high-order neural dynamics. This results in a substantial mismatch between the pretraining objectives and the downstream decoding tasks, leading to poor transferability and degraded performance.

---

> > ### Comment · Reviewer_s4VP · 2025-11-24
> > **Response to Questions 1,2,3**
> >
> > For questions 1 and 2, the answer seems complete. Regarding Question 3, have you fine-tuned the foundation models to each task ? And if not, what happens when you do ?

---

> > > ### Author Response · Authors · 2025-11-25
> > >
> > > Regarding Question 3, we performed full fine‑tuning on each downstream task. We sincerely thank the reviewer for the careful reading and insightful feedback, which has helped us improve the clarity and rigor of our manuscript. If you have any further questions, we would be glad to provide further clarification.

---

> ### Author Response · Authors · 2025-11-19
>
> 4. **Question 4:**
>
>    We thank the reviewer for the insightful question regarding the relationship between model size and performance. NeuroSketch-Base and NeuroSketch-Large contain approximately 1.4 M and 4.2 M parameters, respectively. We provide a detailed analysis of their scaling behavior in Appendix F.3.
>
>    As reported in Table 4 of the manuscript, NeuroSketch-Large achieves a clear performance improvement over NeuroSketch-Base on the Du-IN dataset, while their performances are comparable on the other datasets. To further investigate this phenomenon, we performed a subject-level analysis on Du-IN and observed that the larger model consistently outperforms the smaller one across all subjects, though the magnitude of improvement varies. We also plotted the relationship between per-subject accuracy (difficulty) and relative improvement in Figure 6 of the manuscript, which reveals a negative correlation—that is, the lower the baseline accuracy (the more difficult the samples), the greater the gain from model scaling. Notably, for subjects whose NeuroSketch-Base accuracy was below 5%, the larger model achieved relative improvements of over 400%. These findings suggest that scaling up NeuroSketch enhances its capacity to model complex neural dynamics, particularly for challenging subjects or low-signal conditions, while offering limited benefits once the decoding problem becomes relatively simple.
>
> ------
>
> 5. **Question 5:**
>
>    We thank the reviewer for raising this important question regarding the neurophysiological interpretability of our model design. A detailed response can be found in our general reply to all reviewers. A brief summary is provided below:
>
>    (1) To further examine the interpretability of NeuroSketch, we conducted a case study on the Du‑IN dataset using the Score‑CAM [5] method to visualize the model’s spatiotemporal attention patterns. NeuroSketch‑Large was trained on all available channels from two representative subjects, and class‑activation maps were averaged across samples to obtain subject‑specific saliency maps in both spatial and temporal domains. The spatial analysis reveals a strong correspondence between the salient electrodes identified by NeuroSketch and the significant channels reported in the original Du‑IN study[6]. Moreover, these highly activated electrodes are spatially clustered around two cortical regions(vSMC and STG) well established as the core network for speech motor control, demonstrating that NeuroSketch captures biologically meaningful spatial features consistent with known neurophysiology[7-9]. In the temporal domain, the saliency maps exhibit distinct activation peaks during the mid‑trial period, followed by reduced activation toward both early and late segments. This pattern indicates that NeuroSketch focuses on short‑range dependencies rather than allocating attention uniformly over time, which aligns with the transient temporal dynamics of neural signals.
>
>    (2) To evaluate how representational capacity evolves across network depth, we conducted a rank‑based analysis following the method of Yu et al.[10]. Given the superior performance of CNN‑2D‑based models over Transformer‑based observed in Sections 2 and 3, we further examined two representative models: NeuroSketch‑Large (CNN‑2D‑based) and MedFormer (Transformer‑based). For each layer, we computed the $\varepsilon$‑rank ratio of the output representations obtained through singular value decomposition. This measure reflects the proportion of effective rank relative to the maximum possible rank and provides an interpretable indicator of each model’s expressive diversity at different depths. Results show that: (1) The embeddings of multi-channel brain signals exhibit substantially higher ranks than those typically reported for conventional time‑series data [10, 11]. (2) The two architectures differ in how their ranks evolve across layers. MedFormer shows a consistent decline in the $\varepsilon$‑rank ratio with depth. In contrast, NeuroSketch displays a gradual increase in rank, reaching nearly full rank in deeper stages. This pattern suggests that CNN‑2D‑based models are better at compressing neural signals into more diverse and information‑rich representations than Transformer-based models.

---

> > ### Author Response · Authors · 2025-11-19
> >
> > **Citations:**
> >
> > [1] Zhang C, Kim Y K, Eskandarian A. EEG-inception: an accurate and robust end-to-end neural network for EEG-based motor imagery classification[J]. Journal of Neural Engineering, 2021, 18(4): 046014.
> >
> > [2] Barmpas K, Panagakis Y, Adamos D A, et al. BrainWave-Scattering Net: a lightweight network for EEG-based motor imagery recognition[J]. Journal of Neural Engineering, 2023, 20(5): 056014.
> >
> > [3] Jiang W B, Zhao L M, Lu B L. Large brain model for learning generic representations with tremendous EEG data in BCI[J]. arXiv preprint arXiv:2405.18765, 2024.
> >
> > [4] Wang G, Liu W, He Y, et al. Eegpt: Pretrained transformer for universal and reliable representation of eeg signals[J].
> > Advances in Neural Information Processing Systems, 2024, 37: 39249-39280.
> >
> > [5] Wang, H., Wang, Z., Du, M., Yang, F., Zhang, Z., Ding, S., ... & Hu, X. (2020). Score-CAM: Score-weighted visual explanations for convolutional neural networks. In *Proceedings of the IEEE/CVF conference on computer vision and pattern recognition workshops* (pp. 24-25).
> >
> > [6] Zheng, H., Wang, H., Jiang, W., Chen, Z., He, L., Lin, P., ... & Liu, Y. (2024). Du-IN: Discrete units-guided mask modeling for decoding speech from Intracranial Neural signals. *Advances in Neural Information Processing Systems*, *37*, 79996-80033.
> >
> > [7] Bouchard K E, Mesgarani N, Johnson K, et al. Functional organization of human sensorimotor cortex for speech articulation[J]. Nature, 2013, 495(7441): 327-332.
> >
> > [8] Hickok G, Poeppel D. The cortical organization of speech processing[J]. Nature reviews neuroscience, 2007, 8(5): 393-402.
> >
> > [9] Chartier J, Anumanchipalli G K, Johnson K, et al. Encoding of articulatory kinematic trajectories in human speech sensorimotor cortex[J]. Neuron, 2018, 98(5): 1042-1054. e4.
> >
> > [10] Yu A, Maddix D C, Han B, et al. Understanding Transformers for Time Series: Rank Structure, Flow-of-ranks, and Compressibility[J]. arXiv preprint arXiv:2510.03358, 2025.
> >
> > [11] Liang Z, Zhu J, Sun W. Why attention fails: The degeneration of transformers into mlps in time series forecasting[J]. arXiv preprint arXiv:2509.20942, 2025.

---

> > ### Comment · Reviewer_s4VP · 2025-11-24
> > **Response to Questions 4, 5**
> >
> > Thank you for including these results. These make the paper stronger.

---

### Official Review · Reviewer_jvHa · 2025-10-30

**Soundness:** 2
**Presentation:** 2
**Contribution:** 2
**Rating:** 4
**Confidence:** 4

**Summary:**

This paper introduces NeuroSketch, a neural decoding framework that enhances the performance of decoding multiple types of brain signals (EEG, SEEG, ECoG) through systematic architectural optimization—from basic architecture selection to macro- and micro-level structural improvements. The authors conducted over 5,000 experiments across eight tasks (covering visual, auditory, and speech modalities), demonstrating that NeuroSketch achieves state-of-the-art (SOTA) performance on multiple benchmarks. The core contribution of this framework lies in its ability to model the spatiotemporal characteristics of brain signals, with step-by-step optimizations showing consistent performance gains.

**Strengths:**

1. Systematic Architectural Exploration with a Clear Optimization Path:
The paper begins with a comparison of basic architectures (CNN-1D/2D, GRU, Transformer, etc.) and progressively delves into macro (latent space transformation) and micro (convolution operation optimization) levels of design. This forms a complete and logically rigorous optimization roadmap, offering high interpretability and methodological value.

2. Large-Scale, Multi-Modal Experimental Validation:
Extensive validation was performed across three modalities (visual, auditory, speech), three brain signal types (EEG, SEEG, ECoG), and eight tasks. The experimental scale is substantial (over 5,000 experiments), making the results highly credible and demonstrating strong generalization capability.

3. Tailored Modeling of Brain Signal Characteristics:
The work explicitly addresses the transient temporal dynamics and spatial locality inherent in neural decoding tasks. Designs such as CNN-2D, group convolution, and early downsampling (Pagoda approach) effectively capture these characteristics, reflecting a deep understanding of the nature of neural signals.

4. Effective Balance Between Computational Efficiency and Performance:
The optimization process considers not only performance improvement but also computational cost (e.g., GFLOPs comparison). Proposed strategies like the Step approach, Pagoda approach, and Group Convolution significantly reduce computational burden while maintaining or even improving performance.

**Weaknesses:**

1. Lack of Discussion on Neurophysiological Interpretability:
Although the model performs excellently, the paper does not deeply analyze whether the neural representations learned by NeuroSketch are interpretable from a neuroscience perspective (e.g., correspondence to brain region activation or cognitive processes). This is an important dimension in BCI research.

2. Insufficient Comparison with Some Existing Neural Decoding-Specific Models:
While comparisons are made against several general time-series models and some brain-specific models, the comparison with certain recent architectures specifically designed for EEG/SEEG [1,2] is not comprehensive enough. This might fail to fully demonstrate its advantages over the best methods in the field.

3. Insufficient Exploration of Multi-Modal Fusion and Cross-Modal Generalization:
Although tested on multiple modalities, the paper does not explore the model's generalization ability across modalities (e.g., transferring a model trained on visual tasks to auditory tasks), nor does it attempt multi-modal fusion decoding, which holds significant value for future BCI systems.

4. Inadequate Analysis of Individual Differences and Cross-Subject Generalization:
Although experiments were conducted on data from different subjects, the systematic analysis of the model's ability to handle individual differences and its cross-subject decoding capability is relatively limited. Results for cross-subject unified training or adaptation strategies are not provided.

5. Ablation Studies are Not Comprehensive Enough:
Although the step-by-step optimization process is presented, a systematic ablation study on the independent contribution of each component in the final NeuroSketch model (e.g., GeM pooling, residual connections) is lacking. This makes it difficult to judge the specific impact of each part on the final performance.

**References:**

[1] Singh, A., Thomas, T., Li, J., Hickok, G., Pitkow, X., & Tandon, N. (2025). Transfer learning via distributed brain recordings enables reliable speech decoding. *Nature Communications, 16*(1), 8749.

[2] Chen, X., Wang, R., Khalilian-Gourtani, A., Yu, L., Dugan, P., Friedman, D., ... & Flinker, A. (2024). A neural speech decoding framework leveraging deep learning and speech synthesis. *Nature Machine Intelligence, 6*(4), 467-480.

**Questions:**

See Weaknesses.

**Details Of Ethics Concerns:**

No ethics concerns are apparent. The public datasets are used appropriately.

---

> ### Author Response · Authors · 2025-11-19
>
> We sincerely thank the reviewer for the constructive feedback and valuable suggestions, which have helped us improve the clarity, completeness, and broader impact of our work. We address each identified weakness and suggestion in detail below.
>
> 1. **Weaknesses 1:**
>
>    Thank you for the suggestion to include a discussion on the neurophysiological interpretability of our model. A detailed response can be found in Point 1 of our general reply to all reviewers. A brief summary is provided below:
>
>    To further examine the interpretability of NeuroSketch, we conducted a case study on the Du‑IN dataset using the Score‑CAM [1] method to visualize the model’s spatiotemporal attention patterns. NeuroSketch‑Large was trained on all available channels from two representative subjects, and class‑activation maps were averaged across samples to obtain subject‑specific saliency maps in both spatial and temporal domains. The spatial analysis reveals a strong correspondence between the salient electrodes identified by NeuroSketch and the significant channels reported in the original Du‑IN study[2]. Moreover, these highly activated electrodes are spatially clustered around two cortical regions(vSMC and STG) well established as the core network for speech motor control, demonstrating that NeuroSketch captures biologically meaningful spatial features consistent with known neurophysiology[3-5]. In the temporal domain, the saliency maps exhibit distinct activation peaks during the mid‑trial period, followed by reduced activation toward both early and late segments. This pattern indicates that NeuroSketch focuses on short‑range dependencies rather than allocating attention uniformly over time, which aligns with the transient temporal dynamics of neural signals.
>
>    We have included these analyses and results in Appendix F.1 of the revised manuscript, and we hope that they help clarify and address your concern.
>
> ------

---

> > ### Author Response · Authors · 2025-11-19
> >
> > 2. **Weakness 2:**
> >
> > We sincerely thank the reviewer for highlighting the importance of comparing with recent neural decoding–specific architectures, particularly those designed for EEG/SEEG signals [6,7]. For the first cited work [6], which employs a Seq2Seq architecture for phoneme-level speech decoding, we reproduced its CNN–LSTM encoder on our speech decoding datasets Du-IN and Chisco for a fair comparison with NeuroSketch-Base and NeuroSketch-Large. The results(accuracy) are summarized below:
> >
> > | Model                           | Du-IN                 | Chisco-R              | Chisco-I              |
> > | ------------------------------- | --------------------- | --------------------- | --------------------- |
> > | CNN-LSTM(Original, 0.3M Params) | 0.071 $\pm$ 0.003     | 0.050 $\pm$ 0.000     | 0.050 $\pm$ 0.000     |
> > | CNN-LSTM(4 M Params)            | 0.082 $\pm$ 0.002     | 0.050 $\pm$ 0.000     | 0.050 $\pm$ 0.000     |
> > | NeuroSketch-Base                | 0.472 $\pm$ 0.006     | 0.111 $\pm$ 0.001     | **0.103 $\pm$ 0.002** |
> > | NeuroSketch-Large               | **0.652 $\pm$ 0.002** | **0.116 $\pm$ 0.001** | 0.095 $\pm$ 0.005     |
> >
> > These results clearly demonstrate that the proposed NeuroSketch models consistently outperform the CNN–LSTM baseline across all evaluated datasets. The original CNN–LSTM model [6], containing 0.3 M parameters, is inherently limited in its capacity to capture the neural representations required for complex decoding tasks. To ensure that the observed performance gap is not merely due to differences in model scale, we additionally trained a parameter‑matched variant of CNN–LSTM (≈ 4 M parameters, comparable to NeuroSketch‑Large) by increasing the number of LSTM layers and the hidden dimension. As shown in the table, enlarging the CNN–LSTM model yields only a modest improvement on Du‑IN, while its performance remains far below that of NeuroSketch‑Large. This finding indicates that increasing parameter count alone cannot account for the substantial gains achieved by our framework. We attribute this persistent performance gap to the architectural advantages embedded in NeuroSketch. Its design leverages small‑kernel 2D convolutions to capture short‑range temporal dynamics and localized spatial dependencies, which are consistent with the temporal and spatial characteristics of brain activity.
> >
> > Regarding the second work [7], it employs a 3D ResNet designed specifically for its own dataset, where ECoG signals are represented as 3D spatiotemporal tensors of shape `(batch_size, channels, time, height, width)` and mapped to speech synthesizer parameters. In this context, height and width denote the two‑dimensional spatial layout of electrodes on the cortical surface (e.g., an 8×8 grid). This configuration enables the use of 3D convolutions to jointly capture temporal and spatial correlations within the grid‑structured ECoG recordings. However, this architecture is closely tied to the regular grid electrode placement and speech‑reconstruction task of their proprietary dataset. In our datasets, the ECoG recordings from the FacesHouses dataset lack electrode arrangement information, and the EEG and SEEG datasets were acquired from non‑uniform, irregular electrode layouts without clearly defined spatial dimensions (height, width). Therefore, we did not include this model in our comparison, as its design and data assumptions are incompatible with our study.
> >
> > To further enhance the comparison with recent neural decoding–specific architectures, we have included Du-IN [2] in our extended experiments for the revised manuscript. The Du-IN model achieves state-of-the-art performance in speech decoding through pretraining and fine-tuning on the Du-IN dataset. The results are directly taken from the original publication, as summarized below:
> >
> > | Model             | Dataset | Accuracy          |
> > | :---------------- | :------ | :---------------- |
> > | Du-IN             | Du-IN   | 0.627 ± 0.047     |
> > | NeuroSketch-Large | Du-IN   | **0.652 ± 0.002** |
> >
> > We believe this additional comparison provides a more comprehensive and meaningful evaluation, thereby reinforcing the validity of our experimental results and conclusions.

---

> > > ### Author Response · Authors · 2025-11-19
> > >
> > > 3. **Weakness 3:**
> > >
> > >    We appreciate the reviewer’s insightful comment regarding multi-modal fusion and cross-modal generalization. We fully agree that integrating information across modalities (e.g., visual and auditory) represents an important future direction for brain–computer interface (BCI) systems. However, multi-modal fusion and cross-modal generalization necessitate the implementation of a consistent experimental paradigm across different stimulus modalities. For instance, we first record brain signals while a subject reads a sentence, and then record them again while the subject listens to the corresponding audio, ensuring core experimental settings (e.g., stimulus presentation duration and data acquisition standards) are kept uniform across both modalities. Our current datasets, while encompassing different types of stimuli, were collected from different experimental paradigm, making such cross‑modal alignment infeasible. Nonetheless, we acknowledge that cross‑modal and multi‑modal decoding represent important challenges and opportunities for building more generalized BCI systems. In future studies, we plan to actively design unified experimental paradigms and collect corresponding datasets that enable systematic investigation of these directions.
> > >
> > >    ------
> > >
> > > 4. **Weakness 4:**
> > >
> > >    We thank the reviewer for this valuable observation and fully agree that evaluating how well NeuroSketch generalizes across subjects is essential for demonstrating its robustness and real‑world applicability. To explicitly address this concern, we conducted additional cross‑subject experiments on the SEED‑DV and PhysioNet-MI dataset.
> > >
> > >    In these experiments, we ensured during data partitioning that there was no overlap between subjects, so that the models were evaluated purely on their cross‑subject decoding capability. For benchmarking, we compared our approach with two baselines: CBraMod [8], a widely used pretrained brain foundation model, and EEG‑Conformer [9], an unpretrained EEG architecture. The average classification accuracies and standard deviations across unseen subjects are summarized below.
> > >
> > >    | Model   | PhysioNet-MI  | SEED-DV   |
> > >    | - | - | - |
> > >    | EEG-Conformer  | 0.537 $\pm$ 0.004  | 0.032 $\pm$ 0.005 |
> > >    | CBraMod  | 0.613 $\pm$ 0.010 | 0.025 $\pm$ 0.000 |
> > >    | NeuroSketch-Base | **0.621 $\pm$ 0.008** | **0.044 $\pm$ 0.011** |
> > >
> > >    As shown above, NeuroSketch-Base consistently outperforms the baselines on both datasets. On PhysioNet‑MI, NeuroSketch‑Base achieves the highest average accuracy of  62.1%, slightly surpassing CBraMod, a foundation model pretrained on large‑scale EEG data. On SEED‑DV, NeuroSketch-Base also yields clear improvements, achieving a relative gain of +37.5 % compared with EEG‑Conformer. These results demonstrate that NeuroSketch effectively learns subject‑invariant neural representations that generalize well to unseen participants.
> > >
> > >    ------
> > >
> > > 5. **Weakness 5:**
> > >
> > >    We thank the reviewer for the valuable suggestion to provide a more systematic ablation analysis. In addition to the step‑by‑step optimization results already presented in the paper, we have now conducted component‑wise ablation experiments to examine the independent contributions of key architectural elements in the final NeuroSketch model, including the pooling strategy and residual connections.
> > >
> > >    Specifically, for the pooling component, we compared GeM pooling (used in the final model) with average and max pooling across all datasets using NeuroSketch‑Base. The detailed accuracy results are summarized below:
> > >
> > >    | Pooling Method  | Chisco‑R | Chisco‑I | OpenMIIR‑P | OpenMIIR‑I  | ThingsEEG  |
> > >    | :- | :-- | :- | :--- | :--- | :- |
> > >    | GeM Pooling| **0.111 ± 0.001** | **0.103 ± 0.002** | **0.981 ± 0.010** | **0.970 ± 0.002** | **0.196 ± 0.002** |
> > >    | Average Pooling | 0.103 ± 0.002     | 0.095 ± 0.003    | 0.977 ± 0.013  | 0.969 ± 0.002 | 0.192 ± 0.001 |
> > >    | Max Pooling  | 0.085 ± 0.002     | 0.077 ± 0.006     | 0.972 ± 0.007   | 0.946 ± 0.001| 0.170 ± 0.003  |
> > >
> > >    In addition, to evaluate the effect of residual connections, we compared NeuroSketch‑Base with and without this component:
> > >
> > >    | Residual | Chisco‑R  | Chisco‑I | OpenMIIR‑P | OpenMIIR‑I | ThingsEEG |
> > >    | :- | :- | :-- | :- | :- | :- |
> > >    | w/ | **0.111 ± 0.001** | **0.103 ± 0.002** | **0.981 ± 0.010** | **0.970 ± 0.002** | **0.196 ± 0.002** |
> > >    | w/o| 0.084 ± 0.002  | 0.073 ± 0.006| 0.910 ± 0.009 | 0.923 ± 0.004 | 0.110 ± 0.001|
> > >
> > >    As shown, GeM pooling consistently yields slightly higher accuracy than average pooling and clearly outperforms max pooling across all datasets. Similarly, removing residual connections leads to a notable performance drop, confirming that residual learning plays an essential role in stabilizing model optimization and enhancing representational expressiveness.
> > >
> > >    We will include these comprehensive ablation results, together with a concise discussion of the observed trends, in the revised manuscript.
> > >
> > >    ------

---

> > > > ### Author Response · Authors · 2025-11-19
> > > >
> > > > **Citations:**
> > > >
> > > > [1] Wang, H., Wang, Z., Du, M., Yang, F., Zhang, Z., Ding, S., ... & Hu, X. (2020). Score-CAM: Score-weighted visual explanations for convolutional neural networks. In *Proceedings of the IEEE/CVF conference on computer vision and pattern recognition workshops* (pp. 24-25).
> > > >
> > > > [2] Zheng, H., Wang, H., Jiang, W., Chen, Z., He, L., Lin, P., ... & Liu, Y. (2024). Du-IN: Discrete units-guided mask modeling for decoding speech from Intracranial Neural signals. *Advances in Neural Information Processing Systems*, *37*, 79996-80033.
> > > >
> > > > [3] Bouchard K E, Mesgarani N, Johnson K, et al. Functional organization of human sensorimotor cortex for speech articulation[J]. Nature, 2013, 495(7441): 327-332.
> > > >
> > > > [4] Hickok G, Poeppel D. The cortical organization of speech processing[J]. Nature reviews neuroscience, 2007, 8(5): 393-402.
> > > >
> > > > [5] Chartier J, Anumanchipalli G K, Johnson K, et al. Encoding of articulatory kinematic trajectories in human speech sensorimotor cortex[J]. Neuron, 2018, 98(5): 1042-1054. e4.
> > > >
> > > > [6] Singh, A., Thomas, T., Li, J., Hickok, G., Pitkow, X., & Tandon, N. (2025). Transfer learning via distributed brain recordings enables reliable speech decoding. *Nature Communications, 16*(1), 8749.
> > > >
> > > > [7] Chen, X., Wang, R., Khalilian-Gourtani, A., Yu, L., Dugan, P., Friedman, D., ... & Flinker, A. (2024). A neural speech decoding framework leveraging deep learning and speech synthesis. *Nature Machine Intelligence, 6*(4), 467-480.
> > > >
> > > > [8] Wang J, Zhao S, Luo Z, et al. Cbramod: A criss-cross brain foundation model for eeg decoding[J]. arXiv preprint arXiv:2412.07236, 2024.
> > > >
> > > > [9] Song Y, Zheng Q, Liu B, et al. EEG conformer: Convolutional transformer for EEG decoding and visualization[J]. IEEE Transactions on Neural Systems and Rehabilitation Engineering, 2022, 31: 710-719.

---

### Author Response · Authors · 2025-11-19
**To All Reviewers: Additional Analyses of Interpretability 2**

(2) Representation Analysis of CNN-2D and Transformer‑Based Architectures

To investigate this, we examined the model representations from a rank‑based perspective, following the framework proposed by Yu et al.[6] The rank of a feature representation reflects its expressive capacity: a higher rank indicates richer and more diverse features, while a lower rank implies excessive compression and loss of independent information. In a well‑organized hierarchy, the rank is expected to remain high or increase with depth, as deeper layers capture more abstract yet independent features. In our evaluation, since the CNN-2D-based model and the Transformer-based model achieved the overall best and worst performance, respectively, we conducted an analysis on NeuroSketch-Large (CNN-2D-based) and MedFormer (Transformer-based) to examine how the rank of the feature representations produced by each layer evolves across network depth.

For any layer of a given model, let the output representation of a sample be denoted as $\mathbf{O}\in \mathbb{R}^{s\times d}$,  where $s$ represents the sequence length and $d$ the feature dimension. We performed singular value decomposition (SVD) on $\mathbf{O}$ as $\mathbf{O}=\mathbf{U}\Sigma \mathbf{V}^{T}$,  where $\mathbf{U}$ and $\mathbf{V}$ are orthogonal matrices, and $\Sigma$ is a diagonal matrix whose diagonal entries $\sigma _1\ge \sigma _2\ge \cdots \ge \sigma _{\min(s,d)}$ are the singular values. Although the algebraic rank, i.e., the number of nonzero singular values, serves as a strict measure of rank, real‑world data are often noisy. Therefore, we adopt a more practical numerical rank. For a tolerance $\varepsilon>0$, the $\varepsilon$-rank of $\mathbf{O}$ is defined as the number of singular values that are significant relative to the largest one. Specifically, only singular values $\sigma$ that satisfy $\sigma / \sigma_0 > \varepsilon$ are counted as significant. Since the maximum rank (defined as $\min(s, d)$) varies across layers depending on $s$ and $d$, we further compute the $\varepsilon$-rank ratio, defined as the $\varepsilon$-rank relative to the maximum rank. This metric allows for a more intuitive comparison of how the effective rank evolves across layers.

For MedFormer, we extracted the $\varepsilon$-rank ratios of the embedding layer and each of its six encoder layers under three different patch lengths (5, 10, and 20) to examine how patch length affects representational rank. For NeuroSketch‑Large, we computed the $\varepsilon$-rank ratios of the outputs of its three stem layers and all layers across the four feature‑extraction stages, transforming the height–width dimensions of each feature map into a single sequence length for a consistent rank analysis. From the two models' evolution of the $\varepsilon$-rank ratios, we can observe that (1) for multichannel neural signals, the input embeddings (the initial embeddings mapped from raw input signals) are not as low‑rank as those typically observed in some general time‑series foundation models [1, 2].  Specifically, for both MedFormer and NeuroSketch‑Large, when $\varepsilon \le 10^{-2.5}$, the $\varepsilon$-rank ratio of the input embeddings remain above 0.6, indicating that the input embeddings do not collapse into a low‑rank subspace; (2) for both NeuroSketch and MedFormer, we observe distinct trends in how the $\varepsilon$-rank ratio evolves across layers. As the network depth increases, MedFormer tends to progressively reduce the rank of its feature representations, indicating excessive compression of the input information into a lower‑dimensional subspace. In contrast, NeuroSketch exhibits the opposite trend: the rank of its representations gradually increases, approaching full rank in deeper layers. These contrasting trends imply that the two architectures differ fundamentally in how they organize their representation spaces. In neural decoding tasks, compared with the transformer architecture, the CNN-2D architecture is able to compress more robust and information-rich representations.

Citations:

[1] Yu A, Maddix D C, Han B, et al. Understanding Transformers for Time Series: Rank Structure, Flow-of-ranks, and Compressibility[J]. arXiv preprint arXiv:2510.03358, 2025.

[2] Liang Z, Zhu J, Sun W. Why attention fails: The degeneration of transformers into mlps in time series forecasting[J]. arXiv preprint arXiv:2509.20942, 2025.

---

### Author Response · Authors · 2025-11-19
**To All Reviewers: Additional Analyses of Interpretability 1**

We sincerely thank all three reviewers for their thoughtful and valuable feedback. We notice that all reviewers raised concerns regarding the interpretability of our work. In response, we have conducted two additional experiments to enhance the interpretability of our model and results. We have included these analyses and results in Appendix F.1 and F.2 of the revised manuscript, and we hope that they help clarify and address your concern. The detailed descriptions of these two experiments are provided below.

(1) Neurophysiological Interpretability

To further investigate the spatiotemporal features captured by NeuroSketch, we conducted a case study on the Du‑IN dataset[1]. In this analysis, we trained NeuroSketch‑Large on all 115 channels from subject 02 and 90 channels from subject 11, and applied the Score‑CAM [2] method to visualize the model’s decision‑making process across both spatial and temporal dimensions.

Score‑CAM generates class‑activation maps by computing the gradient‑free importance of each feature map and projecting it back to the input space, thereby highlighting which spatiotemporal regions most strongly influence the predicted class. For each sample, we obtained a saliency map and subsequently averaged these maps across all samples from subject 02 and subject 11, respectively, to produce subject‑wise saliency maps. These aggregated maps(Figure 4 in Appendix F.1) were then used to identify regions of interest (ROIs) within both the spatial and temporal domains.

For the temporal analysis, the model exhibits a pronounced activation hotspot concentrated around the mid‑trial period, approximately between 800–1600 ms for subject 11 and 1200–1800 ms for subject 02, followed by a gradual decrease in saliency toward both the early and late segments. This pattern indicates that NeuroSketch focuses on short‑range dependencies rather than allocating attention uniformly over time, which aligns with the transient temporal dynamics of neural signals.

For the spatial analysis, we averaged each subject‑wise saliency map across the temporal dimension to obtain a saliency score for every channel. Notably, for subject 11, the channels previously reported as significant in the original study [1] were 52, 55, 56, 57, 65, 74, 75, 76, 77, 78. Except for channel 65, all nine of these channels appeared within our top 20 most salient channels identified by NeuroSketch. Similarly, for subject 02, the significant channels reported in [1] were 72, 73, 74, 75, 76, 77, 100, 109, 110, 111, among which channels 72–77 were also ranked within our top 20. Beyond these specific electrodes, the remaining highly salient channels detected by our model were found to be spatially clustered around these previously reported regions, indicating that NeuroSketch captures spatial activity patterns that are consistent with experimentally validated cortical regions.

More importantly, the top‑contributing electrodes identified by NeuroSketch are located within or near the ventral sensorimotor cortex (vSMC) and the bilateral superior temporal gyrus (STG), two cortical regions well established as the core network for speech motor control[3-5]. The model’s focus on these regions highlights its ability to capture biologically interpretable and functionally grounded spatial activation patterns, further validating the neurophysiological relevance of our saliency findings.

Citations:

[1] Zheng, H., Wang, H., Jiang, W., Chen, Z., He, L., Lin, P., ... & Liu, Y. (2024). Du-IN: Discrete units-guided mask modeling for decoding speech from Intracranial Neural signals. *Advances in Neural Information Processing Systems*, *37*, 79996-80033.

[2] Wang, H., Wang, Z., Du, M., Yang, F., Zhang, Z., Ding, S., ... & Hu, X. (2020). Score-CAM: Score-weighted visual explanations for convolutional neural networks. In *Proceedings of the IEEE/CVF conference on computer vision and pattern recognition workshops* (pp. 24-25).

[3] Bouchard K E, Mesgarani N, Johnson K, et al. Functional organization of human sensorimotor cortex for speech articulation[J]. Nature, 2013, 495(7441): 327-332.

[4] Hickok G, Poeppel D. The cortical organization of speech processing[J]. Nature reviews neuroscience, 2007, 8(5): 393-402.

[5] Chartier J, Anumanchipalli G K, Johnson K, et al. Encoding of articulatory kinematic trajectories in human speech sensorimotor cortex[J]. Neuron, 2018, 98(5): 1042-1054. e4.

---

### Meta-Review · Area_Chair_9uuf · 2025-12-26

**Summary:**

This paper introduces NeuroSketch—a technique to build AI models that translate brain signals into useful information. Instead of inventing new math threories or techniques to clean up the signals, the authors focused on carefully testing and improving the basic structure of the model. The authors first compared common AI building blocks (like different types of neural networks), and found that 2D CNN works best for brain data. Then they kept improving that style step by step, i.e., changing how the model processes information inside, what kind of filters it uses, how it summarizes data, whether to add shortcuts, and so on.

The authors ran over 5,000 experiments using brain data from three different types (EEG, sEEG, ECoG) across visual, sound, and speech tasks. In almost all cases, NeuroSketch did better than existing methods. The reviewers said this is an engineering work and it lacks comparisons against state-of-the-art works. The reviewers believe the authors did not reimplement prior works well, and they did not explain why  CNN‑2D works better. The authors responded with a lot of new results, i.e., they added new tests, and more comparisons. However, the reviewers still doubted the novelty of this paper and no one raised their scores.

**Reviewer Concerns:**

Main Concerns from Multiple Reviewers:

1. Does the model understand the brain data? Reviewers questioned if NeuroSketch's great results come from real brain patterns or if it's just memorizing data without any real insight. Answer: The authors added tests that show which parts of the brain signals the model pays attention to. The results matched what brain scientists already know.

2. Did they compare it fairly to other models? Some reviewers felt important existing methods were missing or not implemented correctly. Answer: The authors explained that they used the official code for all comparisons. They also added many new ones, including popular brain-specific models and newer "foundation" models. **However, the implementations of prior works presented by the authors cannot achieve the good results in their orginal papers**. Therefore, the reviewers remained negative about this.

3. Does it work well on cross-subject generalization? For brain-computer interfaces, the model needs to work on different people without retraining. Answer: The authors ran extra tests on new datasets and showed that NeuroSketch handles unseen people well and beats existing methods, including one big pretrained brain model.

4. Which parts of the model actually help the most? Reviewers wanted clearer proof of which design choices really work.
Answer: The authors added ablations showing that certain choices, each improve results on their own.

Reviewer-Specific Questions:

1. Reviewer jvHa asked about interpretability, missing comparisons, cross-modal work, generalization to new people, and more detailed breakdowns of the model parts. The authors added new interpretability studies, more comparisons, tests on new subjects, and clear breakdowns.

2. Reviewer s4VP asked about model size, how much data it needs, comparisons to smaller models and big foundation models, and whether the choices make sense biologically. The authors added more comparisons, looked at how performance changes with size, and showed brain-based evidence for why their design works.

3. Reviewer q1SR questioned how baselines were built, why they used 2D CNNs, and whether this is just engineering. The authors explained their choices, showed why 2D CNNs are a good fit for brain signals (they handle local patterns well), and added extra analysis to explain why CNNs beat Transformers.

**I do not think the authors answer the questions from reviewer q1SR well, because the authors did not explain why they cannot repeat the results achieved by prior works**.

**Reviewer Scores:**

I think the scores from reviewers are reasonable.

---

### Decision · Program_Chairs · 2026-01-26

Reject